

# Meteorological ingredients of heavy precipitation and subsequent lake filling episodes in the northwestern Sahara

Joëlle C. Rieder[1], Franziska Aemisegger[1], Elad Dente[2,3], and Moshe Armon[1]

[1]Institute for Atmospheric and Climate Science, ETH Zurich, 8092 Zurich, Switzerland
[2]Department of Geology and Environmental Science, University of Pittsburgh, PA, USA
[3]School of Environmental Sciences, University of Haifa, Israel

**Correspondence:** Moshe Armon (moshe.armon@env.ethz.ch)

**Abstract.** The dry Sahara was potentially wetter in the past during the warm African Humid Period. Although debated, this climatic shift is a possible scenario in a future warmer climate. One major line of evidence reported for past green periods in the Sahara is the presence of paleo-lakes. Even today, Saharan desert lakes get filled from time to time. However, very little is known about these events due to the lack of available in-situ observations. In addition, the hydrometeorological conditions associated with these events have never been systematically investigated. This study proposes to fill this knowledge gap by examining the meteorology of lake-filling episodes (LFEs) of Sebkha el Melah – a commonly dry lake in the northwestern Sahara. Heavy precipitation events (HPEs) and LFEs are identified using a combination of precipitation observations and lake volume estimates derived from satellite remote sensing. Weather reanalysis data is used together with three-dimensional trajectory calculations to investigate the moisture sources and characteristics of weather systems that lead to HPEs and to assess the conditions necessary for producing LFEs. Results show that hundreds of HPEs occurred between 2000 and 2021, but only 6 LFEs eventuate. The ratio between the increase in lake water volume during LFEs and precipitation volume during the HPEs that triggered the lake-filling, known as the runoff coefficient, provides a very useful characteristic to assess storm impacts on water availability. For the 6 LFEs investigated in this study, the runoff coefficient ranges across five orders of magnitude and is much smaller than the figures often cited in the literature for the Sahara. We find that LFEs are generated most frequently in autumn by the most intense HPEs, for which the key ingredients are (i) the formation of surface extratropical cyclones to the west of the Atlantic Sahara coastline in interplay with upper-level troughs and lows, (ii) moisture convergence from the tropics and the extratropical North Atlantic, (iii) a premoistening of the region upstream of the catchment over the Sahara through a recycling-domino-process, (iv) coupled or sequential lifting processes (e.g., orographic lifting and large-scale forcing), and (v) the stationarity of synoptic systems that result in long-duration (typically 3 d) HPEs. Based on the insights gained into Saharan LFEs in the present-day climate, we suggest that the initial filling and persistence of Saharan lakes may be related to changes in the intensity and frequency of HPEs, rather than a change to mean precipitation alone. Future studies can leverage these insights to better assess the mechanisms involved in the greening of the Sahara in the past and also in a warmer future.





# 1 Introduction

## 1.1 Saharan lakes in the past, present, and future

Inundated lakes in the Sahara are presently a rare, mostly undocumented, transient phenomenon. As a consequence of the lack of rainfall (Nicholson, 2011; Morin et al., 2020; Armon et al., 2024) the extent of surface water bodies and vegetation in the Sahara is limited and it is considered mostly uninhabitable. Yet, ample geological evidence indicates that during a relatively warm period in the mid-Holocene, called the African Humid Period (AHP), the Sahara used to be wetter and greener (e.g., Hoelzmann et al., 2000). Although debated, one major line of evidence for a wetter Sahara during this, as well as previous

AHPs, is the reported presence of Saharan (mega-) lakes (e.g., COHMAPMembers, 1988; Hoelzmann et al., 2000; Lézine et al., 2011; Abafoni et al., 2014). Such lakes, or even smaller-scale wetlands rather than megalakes, as was recently suggested by Quade et al. (2018), point to increased precipitation during such AHPs. When these lakes get filled, they can alter the regional atmospheric circulation and enhance precipitation, therefore locally triggering positive land surface feedbacks (e.g., Krinner et al., 2012; Chandan and Peltier, 2020; Specht et al., 2022). However, currently, the mechanisms involved in both the initial

filling and persistence of Saharan lakes are de facto unclear.

The interest in the processes leading to lake-filling episodes (LFEs) is raised not only because of paleoenvironmental proxies indicating wetter conditions but also given current climate warming. While in many regions present-day anthropogenically-induced climate change decreases water availability (e.g., Archer and Predick, 2008; Cissé et al., 2022), intriguingly, some desert regions may exhibit increased water availability with climate warming. For example, in the Indian Deserts, boosting

food production was recently linked to climate-change-induced modifications of the hydrological regime due to a rise in precipitation (Rajesh and Goswami, 2023). For the Sahara, climate model projections show a pronounced increase in precipitation throughout most of the desert (mean annual precipitation increase of 41.5 % with a 4°C warming, at an SSP5-8.5 scenario, compared with the period 1850–1900), and a decrease in precipitation for the northwestern Sahara ($\sim$ -10 % – -30 %; Iturbide et al., 2021; Gutiérrez et al., 2021). Very large uncertainties, of an order of magnitude larger than the projected change, are

associated with these precipitation projections, especially in the northwestern part of the Sahara desert. Thus, the future of surface water resources in the Sahara is unclear.

In deserts, precipitation matters not only as a key component of the freshwater mass balance but also as a potentially high-impact hazard. Despite the vast uninhabited areas, there are several large urban centres bordering the Sahara. Storm Daniel, which affected northern Libya in September 2023 is a recent example of how deadly and devastating heavy precipitation

events (HPEs) can be in a desert area. This storm induced flash floods in ephemeral streams draining south into the Sahara (Copernicus, 2023) and led to the filling of Saharan lakes that have been empty since the start of Landsat observations (i.e., 1984; Pekel et al., 2016). While some studies previously addressed specific HPEs in the Sahara and their link with floods (e.g. Fink and Knippertz, 2003; Schepanski et al., 2012; Zurqani et al., 2022), to the best of our knowledge, only one study thoroughly analysed the climatology of HPEs in this region (Armon et al., 2024), but it does not include the link to LFEs.

The study of recent LFEs in the Sahara is fairly limited and outdated and is mainly based on observations collected by Dubief (1953) until the 1950s.



To better understand the processes involved in such potentially devastating rainstorms, their capacity to fill Saharan lakes in past, present, and future climates, and to clarify the impact of projected precipitation changes on water availability, we need a better quantitative understanding of the relationship between the atmospheric processes triggering rainfall in the Sahara and the desert hydrology leading to lake-filling in the present-day climate.

## 1.2 Atmospheric circulation controls on precipitation in the Sahara and its link with paleolakes

The processes leading to the dryness of the Sahara are generally well known. A year-round subsidence over the Sahara prevents strong convection (e.g., Rodwell and Hoskins, 1996; Roca et al., 2005; Nicholson, 2011). During winter, the subsidence is established by the down-branches of the Hadley and Ferrel cells which induce the subtropical high pressure systems (e.g., Nicholson, 2011). An anticyclonic flow establishes over the Sahara at the surface, creating dry northeasterly trades – the Harmattan (e.g., Santos-Soares, 2015; Dahinden et al., 2021). During summer, the heating of the surface induces a low-level heat low (Fig. 1; Chen, 2005; Nicholson, 2011). Nevertheless, even in summer several factors reinforce the weak subsidence from the Hadley cell, and suppress the ascent of air needed for precipitation formation: (i) the compensating downward motion from the Asian Monsoon, (ii) the ageostrophic descending force at the right exit of the African Easterly Jet, and (iii) the surface cooling at the west coast of the Sahara, through cold coastal upwelling of the Canary Current and maritime stratus cooling (Rodwell and Hoskins, 1996, 2001; Webster and Fasullo, 2003; Chen, 2005; Miyasaka and Nakamura, 2005).

But, given the strong subsidence aloft in all seasons, how can precipitation still occur in the Sahara? Generally, studies focusing on precipitation in the Sahara are scarce, in particular with respect to heavy precipitation (Armon et al., 2024). A few studies have investigated regional-scale precipitation events in the Sahara to identify the synoptic systems which can lead to Saharan precipitation, but they focused mainly on the periphery of the desert (e.g., Nicholson, 1981; Knippertz et al., 2003; Roca et al., 2005; Rubin et al., 2007).

As the Sahara marks the subtropical transition region between extratropical and tropical weather systems, the northern Sahara receives most precipitation during winter from extratropical cyclones or fronts connected to upper-level troughs, dominantly reaching the northern regions of the Sahara, through the equatorward shift of the westerly jet stream (e.g., Nicholson, 1981; Harada et al., 2003; Armon et al., 2024). In the southern Sahara most precipitation falls during summer due to tropical depressions, cloud clusters, or squall lines, originating from African Easterly Waves and potentially developing into mesoscale convective systems (e.g., Nicholson, 1981, 2011) as the ITCZ moves northward (e.g., Nicholson, 1981; Harada et al., 2003). During transition seasons, the interaction of the extratropical upper-level forcing and tropical systems creates precipitation over the Sahara, mainly at the eastern and western boundaries of the desert (e.g., Flohn, 1975; Nicholson, 1981, 2000; Knippertz et al., 2003; Warner, 2004; de Vries, 2021). A prominent feature associated with this rainfall are tropical plumes, i.e., elongated cloud bands stretching from southeast to northwest, or features similar to these tropical moisture excursions (Mcguirk et al., 1987; Knippertz, 2003; Rubin et al., 2007; Nicholson, 2011).

While different systems potentially cause Saharan precipitation, it is unclear whether similar systems are the cause of wetter periods in the past. Various mechanisms have been associated with increased frequency and intensity of precipitation and the filling of lakes in the geological past. The simplest, most discussed one, is the invigoration and poleward migration of the





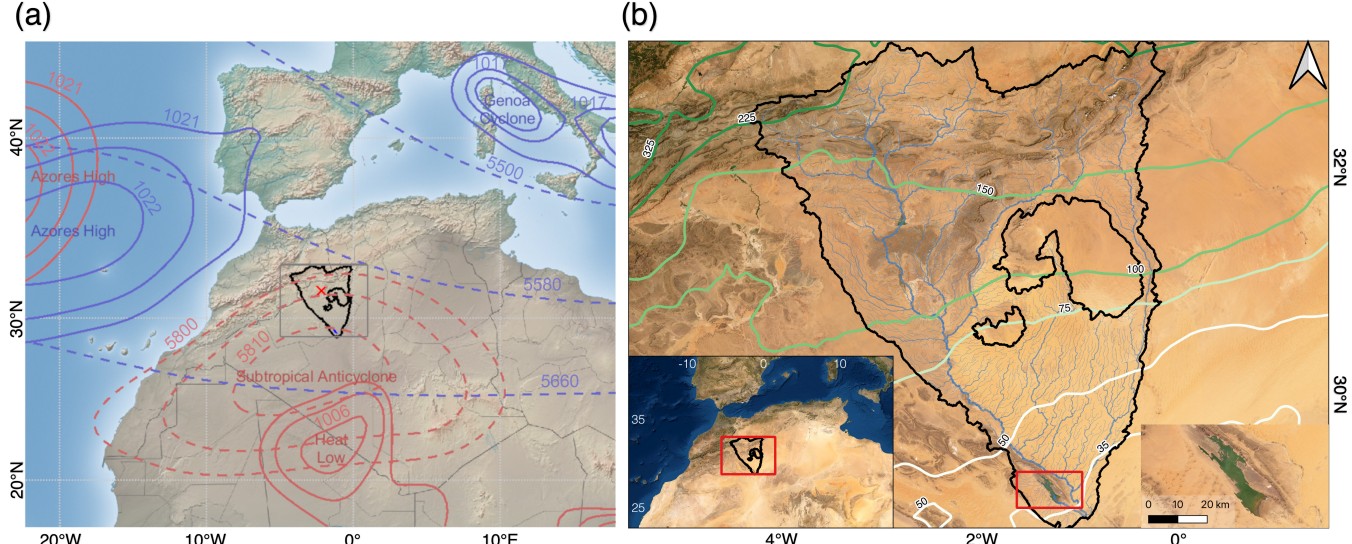

**Figure 1.** Overview of (a) the synoptic climatology over the period 06/2000–05/2021 and the topography (from Natural Earth), and (b) the study region around the Sebkha el Melah catchment (Lehner et al., 2008). In (a), mean winter (blue lines) and summer (red lines) prominent synoptic systems with selected contours of sea level pressure (solid lines; hPa) and mid-troposphere geopotential height (500 hPa, dashed lines; m). Sebkha el Melah catchment is outlined in black. In (b), satellite imagery of the catchment, and of the northwestern Sahara and the lake (small insets) from ESRI world imagery (Earthstar Geographics | Esri, © OpenStreetMap contributors, TomTom, Garmin, FAO, NOAA, USGS), river network (blue lines; Lehner and Grill, 2013), and IMERG precipitation climatology (06/2000–05/2021, green-white shaded isohyets; mm y$^{-1}$).

African Monsoon in response to increased summer insolation during the early Holocene, contributing to increased rainfall throughout the Sahara (e.g., COHMAPMembers, 1988; Lézine et al., 2011). In addition to changes in heating and insolation, increased tropical plume activity was suggested to be associated with larger rainfall input throughout the Sahara during past glacial periods (e.g., Yokochi et al., 2019), based on the isotopic analysis of fossil groundwater (Sonntag et al., 1978; Abouel-

magd et al., 2012), or toward the end of the AHP in the northwestern Sahara (Skinner and Poulsen, 2016). However, even collectively, these mechanisms cannot explain the reported amounts of precipitation needed to maintain proposed Saharan lakes (e.g., Claussen and Gayler, 1997; Quade et al., 2018).

Despite these studies relating atmospheric conditions with lake-filling in the geological past, no study relates present-day lake-filling in the Sahara with atmospheric circulation features. Therefore, this study investigates the conditions associated with

the filling of Sebkha el Melah (Fig. 1) – a normally-dry lake in the northwestern Sahara, filled, on average, only every few years (Mabbutt, 1977; Armon et al., 2020). We combine present-day precipitation and lake-filling remote sensing observations from satellites with meteorological reanalysis data and trajectory-based atmospheric transport diagnostics to explore questions relevant to the hydrometeorology of desert lakes. What are the necessary meteorological ingredients to produce LFEs in the





northwestern Sahara? What triggers the ascent of air, what are the origins of the precipitated moisture and how do these
properties differ between HPEs that trigger LFEs and "normal" HPEs? What portion of rainfall is converted into runoff and is effectively contributing to lake filling? I.e., what is the ratio between catchment precipitation and lake filling volume known as the runoff coefficient (e.g., Merz et al., 2006)? What is the role of below-cloud precipitation evaporation in producing HPEs in the Sahara?

    After presenting the study area (Sect. 2) and methods applied (Sect. 3), we first identify HPEs and LFEs over the catch-
ment area of the lake (Sect. 4.1). Then, we investigate the atmospheric conditions prevailing during an exemplary case of an especially large LFE (Sect. 4.2), and finally take on a climatological perspective highlighting the differences in meteorological conditions associated with LFEs and more common HPEs (Sect. 4.3). In the end, we discuss the ingredients leading to HPEs (Sect. 5.1), the unique characteristics of LFEs (Sect. 5.2), and the relevance of runoff coefficient values to understanding paleo and future climate impacts on surface water availability in the Sahara (Sect. 5.3).

## 115   2   Study area

Sebkha el Melah, sometimes spelt Sabkhat El-Mellah, is located in central-west Algeria (Fig. 1). Its catchment area ($\sim$98,000 km$^2$) lies on the southeastern flank of the High and Saharan Atlas Mountains, with headwaters at $\sim$2000 m elevation and a lowest point (lakefloor) at $\sim$300 m. Mean annual precipitation exhibits a strong gradient, ranging from >200 mm in the north to roughly 30 mm in the south (Fig. 1b). The normally dry lake is fed mainly by Wadi (or Oued) Saoura, a mostly ephemeral stream en-
tering the lake from the southeast (river head in the west). From the confluence of the main tributaries of Wadi Saoura, the Guir and Zousfana streams, Wadi Saoura follows the southeastern edge of the Grand Erg Occidental sand sea. Flowing at the verge of the erg, infiltration losses are so high that during minor to moderate flows water cannot reach as far south as Sebkha el Melah; only major floods (once every few years) are able to reach the lake (Mabbutt, 1977). When the lake reaches full capacity, which only happens in extremely rare floods, water backflows along the same channel through the Messaoud stream south
into the Sahara. Otherwise, the primary source of water loss from the lake is evaporation (Mabbutt, 1977; Armon et al., 2020). While the eastern part of the catchment is covered mainly with highly permeable sands, the northern and western regions are characterised by abundant bedrock outcrops and hamadas, which can contribute much more runoff (e.g., Yair and Kossovsky, 2002; Wheater, 2007). Furthermore, whereas some of the upper part of both the Guir and the Zousfana streams are dammed – Djorf Torba dam in the upper Guir, built in the 1960's (Sarra et al., 2023); and the Sfeissif and Rquiza dams in the upper
Zousfana, built in the 2010's (FAO, 2016) – floods in Wadi Saoura still persist and are able to fill Sebkha el Melah (Armon et al., 2020). Given that minor floods dissipate before reaching the lake, Sebkha el Melah is a highly valuable proxy for heavy precipitation leading to major floods at the Sahara's headwaters.





## 3  Data and Methods

HPEs and LFEs were analysed using three complementary approaches: we (i) identified HPEs using satellite remote sensing
precipitation data, (ii) investigated the meteorological factors leading to HPEs using reanalysis data, and (iii) subsampled LFE-
generating HPEs using a lake-filling remote sensing technique. The different datasets were combined to obtain the best possible
description of the strongest HPEs and LFEs, and alleviate the impact of potential errors by using either of the approaches
exclusively.

### 3.1  Data

### 3.1.1  IMERG remote sensing precipitation

To estimate precipitation amounts we used the sixth version of the Integrated Multi-satellitE Retrievals for Global Precipitation
Measurement (IMERG V06) (Huffman et al., 2020). IMERG data (06/2000 - 05/2021) are half-hourly global rainfall and
snowfall estimations with a grid resolution of $0.1° \times 0.1°$, combining satellite precipitation estimations from multiple sources
(e.g., Huffman et al., n.d.; Hou et al., 2014). These data are currently probably the best high resolution option representing
spatiotemporal rainfall characteristics in the region, as well as in other desert regions (Zambrano-Bigiarini et al., 2017; Islam
et al., 2020; Mahmoud et al., 2021; Zhou et al., 2021; Rachdane et al., 2022; Armon et al., 2024). Still, precipitation estimations
from satellites show potential limitations in the desert, as a result of, e.g., precipitation evaporation (Dinku et al., 2011),
under-detection because of small-scale short-duration precipitation (Li et al., 2021), or miscalibration during numerous inter-
calibration steps with different spatiotemporal resolutions (Huffman et al., 2020).

### 3.1.2  ERA5 reanalysis data

The fifth generation of the European Centre for Medium-range Weather Forecast (ECMWF) ReAnalysis dataset (ERA5) was
used for the meteorological analysis of the HPEs identified with IMERG data. The reanalysis provides hourly output with a
horizontal resolution of $\sim 31$ km (here available on a 0.5° interpolated grid resolution) and with 137 vertical levels, extending
from the surface to 1 hPa (Hersbach et al., 2020). We used a set of common meteorological parameters which enabled us to
inspect the meteorological conditions prevailing during HPEs, including potential vorticity ($PV$). $PV$ is a commonly used
variable which indicates the stability and vorticity (rotational fluid motion) of the atmosphere. It is mostly used to describe the
jet stream waviness and upper-level influence on cyclogenesis as well as destabilisation of the lower levels (e.g., Hoskins et al.,
1985; Portmann et al., 2021). It is important to note that while the resolution of ERA5 is very high compared to other reanalyses,
it is still insufficient to effectively represent convection. Therefore, convective precipitation in the dataset is parameterised,
which may lead to errors in precipitation estimations.





### 3.1.3 Lake observations

To identify the existence and area of water in the lake we used satellite imagery from both Landsat satellite series (Neigh et al., 2022) at 30 m per pixel and the Moderate Resolution Imaging Spectroradiometer (MODIS; Wolfe, 2023) satellites at 250 m per pixel. Since 1984, Landsat satellites have provided the longest continuous imagery of Earth in various wavelength bands.
Previous studies showed that surface water could be detected and mapped using Landsat imagery (e.g. Pekel et al., 2016; Pickens and Sherani, 2020). The revisit time of Landsat satellites in the study area is approximately 16 days. During cloudy periods, which are not frequent in arid areas, consecutive clear images of the lake may be available once a month or even a bit less frequently. To inspect potential LFEs when Landsat imagery was not available, MODIS imagery (available twice per day in daylight since 1999) was used.

## 3.2 Methods

### 3.2.1 Identification of heavy precipitation events

HPEs were identified using the local (pixel-based) climatology of IMERG (Fig. 2a) following Armon et al. (2024). HPEs (Fig. 2d) were defined based on three criteria:

a **Precipitation must locally exceed a climatology-based threshold:** To distinguish between normal and high precipita-
tion intensities we determined the threshold to be the pixel-specific 90% quantile of daily IMERG precipitation (Fig. 2a) conditioned on rainfall occurrence (P90 | P > 1 mm d$^{-1}$). By comparing the daily data to the threshold map, the threshold exceeding grid cells were defined as heavy precipitation for a specific day (Fig. 2b-c).

b **Precipitation must occur over a substantial contiguous area:** Heavy precipitation has to occur over a spatially contin-
uous 'event scale', defined here as $\geq 1000\,\mathrm{km^2}$ – a typical local storm size (e.g., Lohmann et al., 2016; Zoccatelli et al.,
180 2019).

c **The identified precipitation area is spatially and temporally close-to connected (using buffer zones):** We considered a spatial 1-pixel buffer zone surrounding the identified area. If the buffer zones of neighbouring areas were connected, i.e., these areas are distanced 1-2 pixels apart, the areas were considered as one. Similarly, a 2-pixel buffer zone was considered as a temporal buffer zone, connecting precipitation areas of two consecutive daily time steps. It is important
to note that using this approach, if events are small and isolated enough, more than one event can be identified for a given day.

Criteria (b) and (c) were considered to minimise misinterpretations of events due to the occurrence of 'noise' in the dataset as well as heterogeneity in the precipitation threshold.

Rainfall properties during HPE days in the two precipitation datasets, IMERG and ERA5, were compared by integrating
precipitation over the entire catchment. Additionally, the *HPE magnitude*, determined here as the volume of IMERG-based precipitation over the area identified as a HPE was extracted for every HPE. Furthermore, HPEs were categorised into three





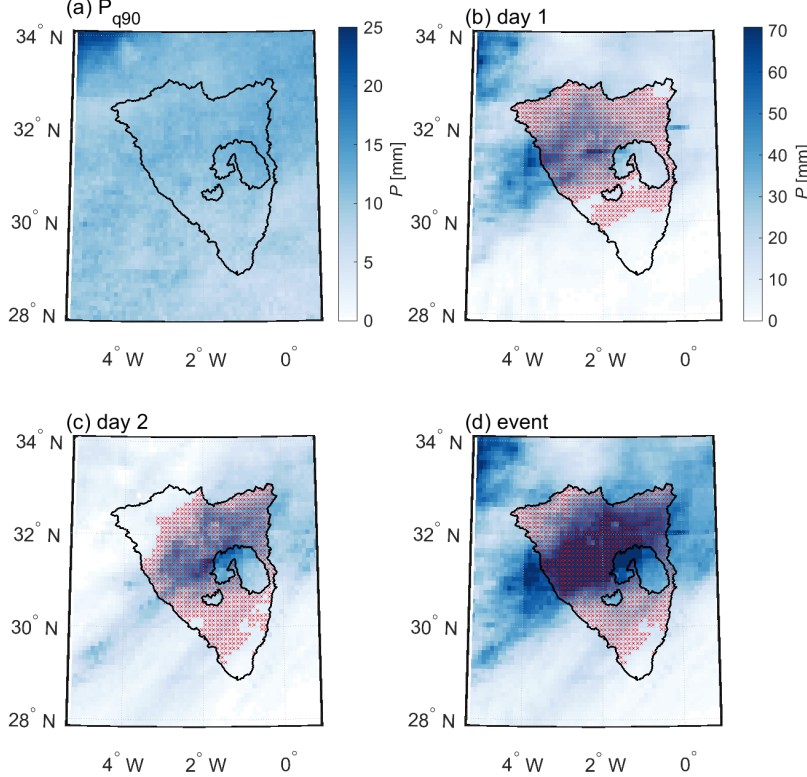

**Figure 2.** Heavy precipitation event identification example for two (28-29/11/2014) out of four days (27-30/11/2024) identified as a HPE. (a) The pixel specific $90^{th}$ quantile of daily precipitation intensity (P90 | P > 1 mm d$^{-1}$). (b) and (c) daily precipitation (shades of blue) for day one and two of this HPE, respectively, and the threshold exceeding grid cells (red). (d) Event accumulated precipitation and threshold exceeding event grid cells. Note that the colorbar has a different amplitude for panel (a) than for panels (b)-(d).

different categories: (i) *LFE-generating HPEs*, (ii) *strong HPEs*, namely those that compose the top 20% of all identified HPEs by their magnitude, and (iii) *medium HPEs*, i.e., all the smaller HPEs.

### 3.2.2 Lake-filling identification and quantification of lake storage changes

A list of LFEs was compiled by identifying time periods in which the lake area increased abruptly, followed by a slow decrease, using a three-step process:

 **1 Landsat-based detection of lake water area:** The full Landsat 5, 7, and 8 surface reflectance cloud-masked imagery was used to detect the existence and extent of water by applying the Modified Normalized Difference Water Index (MNDWI, based on the green and short-wave infrared bands; Xu, 2006; Pekel et al., 2016) for the time period 2000 - 2021 over
200  Sebkha el Melah. Obtaining an accurate MNDWI threshold for water classification depends on local environmental



conditions (e.g., land cover) and the specific Landsat sensor (Pekel et al., 2016). To find a consistent MNDWI threshold over time and across satellite sensors, we performed a sensitivity analysis based on the studied LFEs (Fig. 3a-b). For every LFE maximum-area documentation date, the water area was calculated using various MNDWI thresholds. Then, the desired robust MNDWI threshold was chosen within low-slope parts of all the plots, i.e., where a change of the threshold has the lowest impact on the resulting water areas across the LFEs. Based on this analysis and a manual inspection of the detected water extent, a threshold of 0.4 was chosen to classify water within Sebkha el Melah. (Fig. 3a-b).

2 **LFE validation:** The resulting MNDWI-based lake area time series (Fig. 3c) was used to manually identify LFEs. Because some imagery underestimates the lake area due to cloud cover, partial tile coverage of the lake, void stripes in Landsat 7 data (reducing the area by up to 20 %; Table A1), or other artefacts (Fig. A1), a running maximum was applied to the lake area time series and LFEs were extracted from it. For the maximum lake area within a LFE, we manually chose the closest imagery that captured the full extent of the water. For every LFE we visually inspected the true-colour composite of the Landsat imagery that documented it. The "Observation gap" between LFEs and Landsat-based lake area imagery could be a few weeks. Therefore, evaporation from the lake's surface during the gap period could impact the detection of the maximum wetted area and the derived water volume. We considered here typical evaporation rates for this area of $20\,\mathrm{cm\,month^{-1}}$ (Rognon, 2001; Saggaï and Bachi, 2018). These evaporative losses are expected to affect volume estimates in particular for small LFEs. To not miss any small LFE because of evaporation, we also analyzed the 10 largest HPEs in the study period. These events were examined visually using MODIS's corrected reflectance composite (bands 7-2-1).

3 **Lake water volume and effective runoff coefficients:** To quantify the storage volume corresponding to the lake area detected in steps 1 and 2, we used the hypsometric curve of the lake (Fig. A2) based on the bathymetry model derived by Armon et al. (2020). To compute *effective* runoff coefficients for every LFE, i.e., the coefficients based on the volume of water that eventually filled the lake, we divided the lake storage change between the pre- and post-event conditions by the IMERG-based precipitation volume, taking into account two end-member scenarios: (a) runoff is solely contributed to the lake from precipitation during the specific HPE, and (b) runoff is contributed from precipitation during the HPE as well as precipitation in the following weeks until the lake is observed. For scenario (b), we additionally estimated potential evaporation losses during this time period. A delayed lake stage observation after a HPE artificially changes the effective runoff coefficient. Considering additional rainfall volumes lowers the runoff coefficient, while accounting for evaporation raises it. Based on MODIS observations we know that Sebkha el Melah reaches maximum filling values a few days after HPEs. Therefore, the actual runoff coefficients lie somewhere between these two end-member scenarios.

The main potential drawback in calculating event-based runoff coefficients is the anthropogenically altered hydrology of the catchment, mainly by the construction of dams (Sect. 2). Because the operation schemes of the dams are not publicly available, their specific impact on the result was not calculated. However, the effective runoff coefficient is generally





expected to be lower after the dam construction compared to events before the construction. Other potential sources for
volume and area errors are presented in Supplementary Table A1.

### 3.2.3    Lake volume uncertainty analyses

The potential impacts of the observation gap, the evaporation rate, and the derived volume were computed. Lake area changes
due to evaporation during the observation gap may lead to noticeable deviation of the lake volume associated with the LFE.
Furthermore, additional rainstorms can occur between the filling of the lake and the next Landsat observation. Whereas high-
magnitude HPEs are expected to lead to the filling of the lake, low-magnitude rainstorms are not expected to cause substantial
surface runoff and flooding (e.g.,  Zoccatelli et al., 2019; Shmilovitz et al., 2020). These effects from additional rainfall and the
estimated evaporation impacts are used to present possible uncertainties of the lake volume estimates for each of the LFEs.

Additionally, uncertainties in the volume estimations can rise from the MNDWI thresholds we chose (Fig. 3a). Therefore,
we also present this potential volume uncertainty specifically for each LFE. These uncertainties are further discussed in the
results (Sect. 4.1) in which the rainfall volume estimates from IMERG and the detected lake volumes are compared to derive
runoff coefficient estimates.

### 3.2.4    Trajectory calculations

To characterize the meteorological conditions leading to HPEs and LFEs and identify their moisture sources, we calculated air
parcel backward trajectories. Trajectory calculations from the Sebkha el Melah catchment were initiated every 6 h, computed
10 days backward in time driven by the 3D wind fields from the ERA5 dataset and using the LAGRangian ANalysis TOol
(LAGRANTO; Wernli and Davies, 1997; Sprenger and Wernli, 2015). Trajectories were initiated from 21 different locations
within the catchment, spaced by 70 km. For each of these locations, starting points were selected every 35 hPa in the vertical
within the layer where clouds during HPEs typically form (i.e. between 900 hPa and 200 hPa). This configuration means that
for every HPE 420 trajectories per 6 hourly time step were calculated. In addition, different meteorological variables including
specific humidity ($q$) and relative humidity (RH) were interpolated to the trajectory positions.

To investigate the potential role of precipitation evaporation as well as the injection of moisture into the air parcel within
convective plumes, the atmospheric conditions above (up to 100 hPa) and below (down to the surface) the trajectories were
extracted from ERA5. The following thermodynamic variables, $q$, RH, the boundary layer height, temperature, as well as the
hydrometeor concentrations were interpolated to the geographical positions of the trajectories along the vertical profiles at an
hourly resolution and were averaged across selected trajectories which formed the main moisture transport pathway in the
lower troposphere.

### 3.2.5    Moisture source diagnostics

The moisture sources for precipitation during the identified HPEs were diagnosed with a well-established Lagrangian moisture
source diagnostic (MSD; Sodemann et al., 2008). This diagnostic has been applied in many atmospheric water cycle studies.





**Figure 3.** Lake-filling identification and quantification of the lake's storage. (a) The sensitivity of lake area (dashed lines; left axis) and volume (solid lines; right axis) to the MNDWI threshold over different LFE maximum filling dates. MNDWI thresholds 0.1, 0.25, 0.4, 0.55, and 0.7 are marked (grey lines). (b) An example of different MNDWI threshold lake areas for LFE3 (07/05/2009). MNDWI thresholds 0.1 (light blue), 0.4 (red), and 0.7 (dark blue) are marked. (c) Time series of the lake's area with a 0.4 MNDWI threshold for the study period (06/2000–05/2021) represented by Landsat observations (blue bars), and a smoothed (running 15 data point maxima) series of lake area (red line). HPE dates corresponding to the identified LFEs are marked as red crosses. * = LFEs where the lake was already partially inundated. Landsat-7 image in panel (b) is courtesy of the U.S. Geological Survey.

More specifically, these studies investigated the tropical and subtropical water cycle and the dynamics of the Saharan heat low in Dahinden et al. (2023), and the North Atlantic trade wind water cycle in Aemisegger et al. (2021) and Villiger and Aemisegger (2023). This trajectory-based diagnostic identifies the evaporative source footprint of precipitating waters by establishing a mass balance of vapour phase humidity in the air parcels contributing to the precipitation event. Here we track hourly changes in $q$ (i.e., $\Delta q$) forward in time along each of the trajectories (Fig. 4a). If $\Delta q$ is positive the air parcel is considered to have taken

up moisture through surface evaporation (event #1 in Fig. 4a) or below cloud rain evaporation (event #3 in Fig. 4a). Negative $\Delta q$ are due to precipitation formation in clouds underway (events #2 and #4 in Fig. 4a). In this case, all previous uptakes are





discounted proportionally to their contribution to the humidity lost in the precipitation event. A $\Delta q$ threshold of $0.001\,\mathrm{g\,kg^{-1}}$ is used to avoid spurious uptakes or losses and neglect numerical noise in $\Delta q$ along the trajectories.

Thus, one air parcel generally has multiple moisture sources along its path. Each source is associated with a weight quan-
tifying its contribution to the water vapour at the point of arrival in the cloud forming precipitation above the catchment. The relative moisture contribution of each individual air parcel to the precipitation at a given 6 hourly time step is determined by its share of the total water vapour content carried by the airstreams feeding the precipitation-bearing cloud system observed in the IMERG data set. This approach was chosen to make sure that the moisture sources are identified for all HPEs detected by IMERG data, even when there is no or only limited precipitation in ERA5 data. In Sodemann et al. (2008) only trajectories with
a negative $\Delta q$ and with a relative humidity of more than 80% are considered as contributing to precipitation. Here, because of the convective nature and complex small-scale structure of precipitation systems in the Sahara, the position and the vertical structure of the cloud system producing precipitation in the Sebkha el Melah catchment is not expected to be necessarily accu-rately represented in ERA5. However, we can safely expect ERA5 to faithfully represent the large-scale transport of moisture into the cloud layer above the catchment. We thus assume that the trajectories arriving in the cloud layer contribute to the ob-
served precipitation proportionally to their water vapour content at arrival. We then use the IMERG precipitation observations integrated over the catchment and over the preceding 6 hours to weight the contribution of each 6 hourly time step to the total precipitation of a given HPE. Finally, moisture sources are integrated over different pre-defined regions (Fig. 4b).

## 4 Results

### 4.1 Overview of identified HPEs and LFEs: occurrence, rainfall, and runoff

During the study period (06/2000–05/2021), 250 HPEs occurred in the catchment of Sebkha el Melah over 356 days in total (Fig. 5a, and Supplementary material) with an average of 1.4 events per month. Events are not distributed evenly throughout the year (Fig. A3): autumn is when events are most frequent (2.6 events month$^{-1}$), followed by spring (1.7 month$^{-1}$), winter (0.9 month$^{-1}$), and summer (0.4 month$^{-1}$). Precipitation during HPEs varies between 0.5 mm and 60.8 mm averaged over the catchment, with a mean value of 6.8 mm (Table 1).

Since in-situ rain measurements in the Sahara are scarce, it is difficult to validate precipitation data from both the satellite and reanalysis datasets. While ERA5-based precipitation is consistently lower compared to IMERG-precipitation (-43%, on average; Table 1), high correlation values between the two datasets ($\rho_{Pearson} = 0.87$) and similar ranking for the events ($\rho_{Spearman} = 0.72$) suggest that both can capture HPEs in the catchment (Table 1). However, because of the coarser resolution and parameterised convection in ERA5 (Sect. 3.1.2), convective rainfall – which is an important contributor to the majority of
HPEs in desert areas (e.g., Sharon, 1972; Morin et al., 2020) – is not well represented in the ERA5 data and precipitation rates are probably too low, as was recently observed over the entire Sahara (Armon et al., 2024).

Rarely, HPEs trigger enough runoff to (partially) fill Sebkha el Melah. During the study period, LFEs occurred only six times, partially filling the lake with water volumes of $\leq 0.8\,\mathrm{km^3}$ amounting to about 90 % of its capacity (Figs. 5b, A2, and Table 2). Clearly, LFEs originate from the largest HPEs; each of the LFEs is associated with at least one of the ten largest HPEs





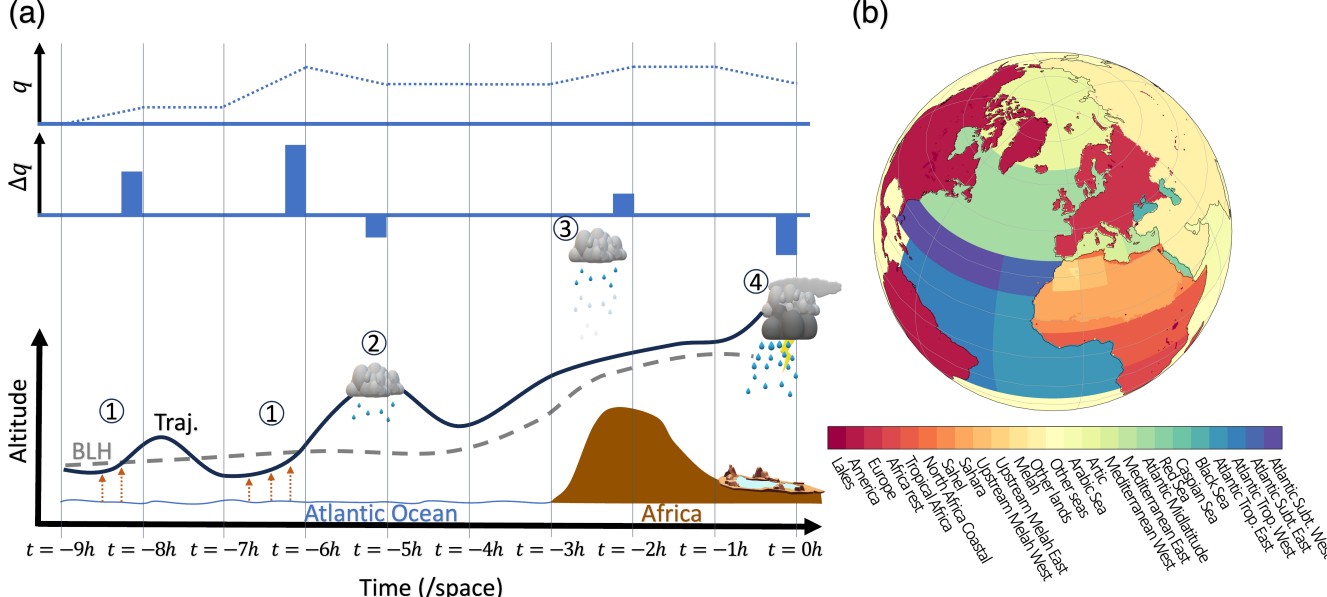

**Figure 4.** Overview of the moisture source diagnostic (MSD). (a) Sketch of an exemplary backward trajectory of an air parcel passing over the Atlantic Ocean on its way to Africa (black line). Time before arrival is given at the bottom. The specific humidity ($q$; blue dashed line) in the air parcel, and its variation ($\Delta q$; blue bars) between consecutive time intervals are in the upper panels. The boundary layer height ($BLH$; grey dashed line) is shown along the trajectory path, as well as moisture increases by (1) surface evaporation/sublimation (orange arrows) and (3) below cloud evaporation/sublimation (rainy cloud above trajectory), and moisture decreases by precipitation (2) along the way and (4) over the catchment. Figure adapted from Sodemann et al. (2008). (b) Moisture source region classification for the MSD.

by HPE magnitude (Sect. 3.2.1), mostly coinciding with longer event durations (typically, 3 d; Fig. 5a). Two of these LFEs resulted each from two sequential HPEs and the other four are a result of single HPEs (Table 2). Hereafter, we number these LFEs chronologically and use the same numbers to describe the corresponding HPEs, such that the first LFE during the study period is termed LFE1 and its associated HPE is termed HPE1. LFE-generating HPEs are characterised by event magnitudes >1.4 km$^3$ (14.7 mm when averaged over the catchment), while strong HPEs have magnitudes of ≥0.66 km$^3$ (6.8 mm) and medium HPEs range between 0.66 km$^3$ and 0.01 km$^3$ (6.7 mm and 0.1 mm, respectively).

While the HPEs that lead to LFEs are characterised by high rainfall values, runoff varies greatly between events. Mean catchment precipitation during these HPEs, when accumulating rainfall during consecutive HPEs if only one LFE eventuated, is between 20 mm and 93 mm. Effective runoff coefficients, which represent the portion of rainfall that actually reaches the lake by surface runoff, are computed for each of the LFEs based on the two end-member scenarios in Sect. 3.2.2-3. Scenario (a), in which precipitation is considered during HPEs only, is represented by the red symbols in Fig. 5b, and scenario (b), in which precipitation is accumulated until the Landsat observation time, is represented by the blue symbols. The calculated effective runoff coefficients of the LFEs vary by five orders of magnitude (<0.001% – ∼10%). While in general higher coefficients



**Table 1.** Comparison of mean catchment precipitation properties during HPEs in IMERG and ERA5.

| Precipitation data source | IMERG | ERA5 |
|---|---|---|
| Mean (standard deviation) | 6.8 mm (8.2 mm) | 3.8 mm (6.3 mm) |
| Median | 4.0 mm | 1.2 mm |
| Bias of ERA5 compared to IMERG | - 43.2 % | |
| Root mean square deviation (RMSD) | 5.1 mm | |
| Pearson correlation ($p$-value) | 0.87 ($\ll 0.01$) | |
| Spearman correlation ($p$-value) | 0.72 ($\ll 0.01$) | |
| Linear regression | $P_{\mathrm{ERA5}} = P_{\mathrm{IMERG}} * 0.7 - 0.6$ mm | |

are exhibited when higher precipitation amounts occur, relatively high coefficient values (>1%) are exhibited in LFE1 (and potentially LFE3; Fig. 5b). Still, the highest effective runoff coefficients are exhibited for the largest events (LFE2 and LFE5), where precipitation reaches nearly 100 mm.

## 4.2 Meteorological conditions during the November 2014 LFE: a case study

To better understand the synoptic ingredients involved in the filling of Sebkha el Melah, we focus now on a specific case study – LFE5 in November 2014 (Fig. 5b, Table 2). This LFE is associated with two consecutive HPEs (#5.1 and 5.2 in Fig. 5a) which combined contributed a volume of 0.32 km$^3$ to the lake, ending a three-year period, during which the lake was almost completely empty, with the exception of a small LFE in October 2012 (LFE4; Fig. 3).

Precipitation during HPE5.1 and HPE5.2 was extremely intense, summing to 93 mm averaged over the catchment, with values locally exceeding 140 mm over the central part of the catchment (Fig. 6). In both subevents of HPE5 two distinct peaks of precipitation appear: HPE5.1 on 22 Nov 2014, and from 23 to 24 Nov 2014, and HPE5.2 on 27 Nov 2014, and from 28 to 29 Nov 2014. Both IMERG- and ERA5-based precipitation are remarkably similar in terms of the timing of precipitation, with ERA5 showing a -22% bias compared to IMERG for catchment mean precipitation, and slightly lower precipitation intensities.

### 4.2.1 Synoptic evolution

During HPE5.1, a low-level cyclone is positioned at the western coast of Morocco, exhibiting increased moisture, and extensive precipitation. This low-level cyclone is formed through two sequential upper-level stratospheric $PV$ streamers, which reach farther south of 30°N and transform into $PV$ cut-offs (Fig. 7a,b). These upper-level $PV$ features persist in the region throughout HPE 5.1. A pronounced anticyclone that extends over eastern North Africa and the Mediterranean forms during the event. The anticyclonic wind field to the east of the study region and the cyclonic wind field to the west induce a convergent, southerly to southwesterly, flow from Equatorial West Africa towards the catchment (Fig. 7a,b). This flow is associated with an elevated moisture band from the tropics, resembling a tropical plume.





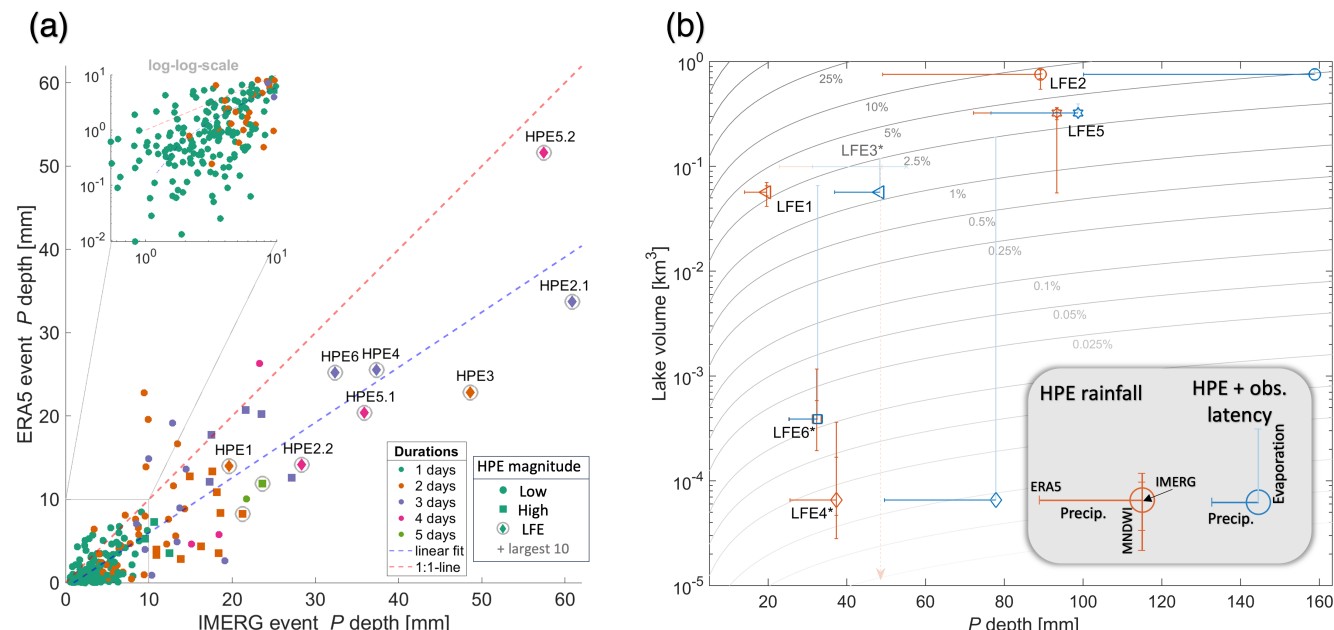

**Figure 5.** Overview of HPEs and LFEs in Sebkha el Melah. (a) Mean catchment precipitation ($P$) depth for IMERG- and ERA5-based data. Event durations in colours, the ten largest HPEs by magnitude are marked with grey circles with LFE-generating HPEs marked with diamonds and labelled, strong HPEs with a magnitude $\geq 80^{\text{th}}$ quantile are marked with squares, and medium HPEs are marked with dots. (b) Precipitation during LFE-generating HPEs compared with the storage change for their associated LFEs. Grey contours are the effective runoff coefficients. Every LFE is shown twice, distinguishing between the two end-member scenarios (Sect. 3.2.2-3), where orange symbols represent rainfall during the HPE only, and the blue symbols represent also the rainfall added until the lake observation (Table 2). Negative horizontal errorbars represent ERA5 precipitation. Vertical orange errorbars show lake storage variability when using MNDWI thresholds other than 0.4 (values are the same as in Fig. 3a). Positive vertical blue errorbars represent evaporative loss estimates. * = LFEs where the lake was already partially inundated. LFE3, shown in semi-transparent colours, is represented only by the evaporative loss scenario, as the clear Landsat observation did not yield additional lake volume (Table 2).

In between HPE5.1 and HPE5.2 (25-26 Nov 2014), the low-level wind in the catchment is weak (not shown). Winds turn
shortly into (north-)easterlies as the upper-level $PV$ cut-off moves to the east, and with it the surface cyclone. Low-level moisture is still high due to the evaporation of previous precipitation and persists in the vicinity of the catchment (Fig. 8a,b). In fact, relative and specific humidity remain high within the boundary layer throughout HPE5, with values between 60 % and 100 %, and $> 7 \, \text{g kg}^{-1}$, respectively. With the development of an upper-level stratospheric $PV$ streamer on 26 Nov 2014, a cyclone formed at the Iberian west coast (not shown). Thereafter, the streamer extends down to 25°N.
In the initial stage of HPE5.2 (27 Nov 2014), the narrow $PV$ streamer moves over the catchment, supporting the ascent of the (remaining) moisture in the vicinity of the catchment and producing light precipitation. At the same time, a deep cyclone



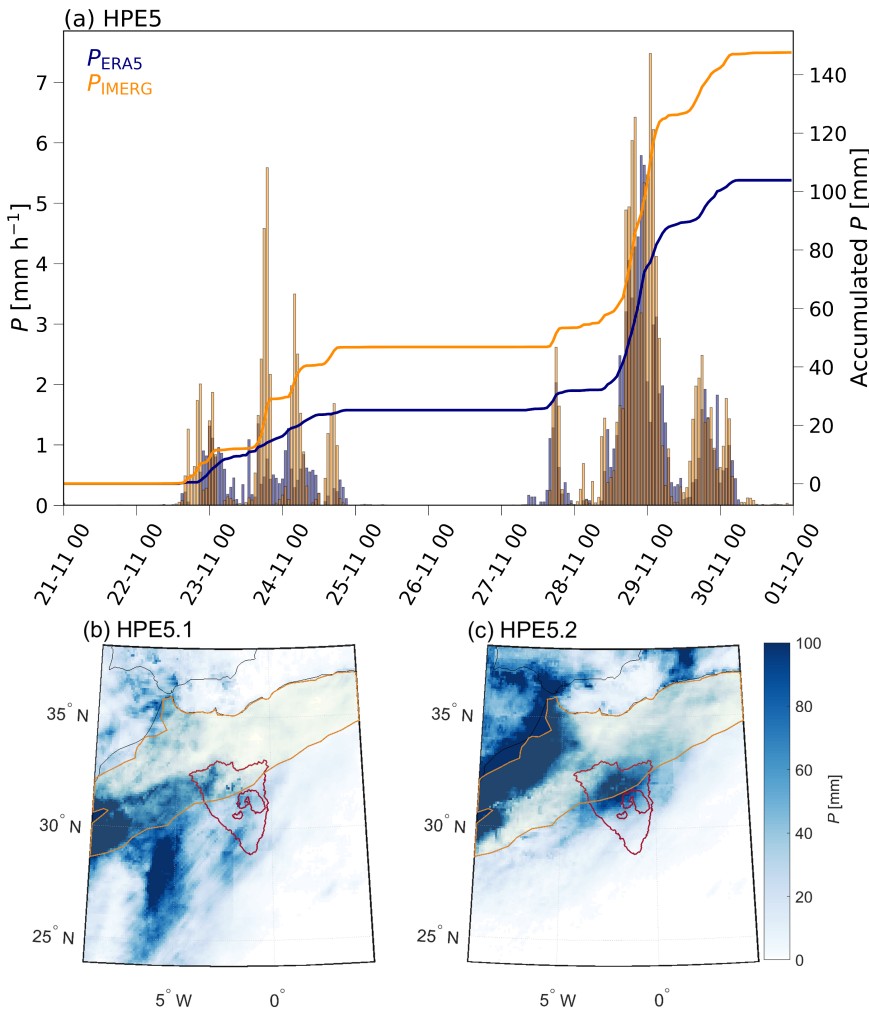

**Figure 6.** Precipitation during HPE5. (a) Hourly evolution of precipitation intensity (left axis; bars) and accumulated precipitation (right axis; lines) for IMERG (orange) and ERA5 (blue) data interpolated to the central part of the catchment, at the city of Béchar (Fig. 1b). (b) Accumulated IMERG-based precipitation during HPE5.1 (21 to 24 Nov 2014). (c) Similar to (b) but for HPE5.2 (27 to 30 Nov 2014). Red contour is the Sebkha el Melah catchment, brown shading denotes the Atlas Mountain Range (contour based on NaturalEarth, 2023).

in the north starts to move southward as a large upper-level stratospheric $PV$ cut-off is emerging, approaching or continuously reforming from the southern tip of Greenland, over the Iberian Peninsula, to the western coast of Morocco (Fig. 7c). Moisture accumulates in front of this cyclone, associated also with an extratropical frontal system (Fig. A4). When the cyclone reaches the western coast of Morocco, the low-level southwesterlies bypass the Atlas Mountains, and bring moisture far into the Sahara (Fig. 7c,d). On 29 to 30 Nov 2014, the cold front of the low-level cyclone and the southern part of the upper-level $PV$ cut-off move slowly over the catchment. This brings catchment precipitation to its peak (Fig. 6a).





Throughout HPE5 isentropes are weakly stratified across the troposphere, indicating potentially strong convection (Fig. 8b,c). Thus the two events are accompanied by a low-level destabilisation, coinciding roughly with the peaks in precipitation.

The approach of the upper-level $PV$ streamers and cut-offs contribute to the destabilisation and initialisation of precipitation upstream of the catchment. The upper-level $PV$ structures, however, do not cross the catchment during precipitation peaks, except for 27 Nov 2014, thus their role in precipitation formation is mainly at the large scale, as triggers of the moist low-level southerly flow towards the Atlas, enabling intense convection over the orography. The modelled hydrometeor concentrations during HPE5 indicate that deep mixed-phase precipitating clouds, reaching cloud tops higher than 300 hPa, characterised this

event (Fig. 8c), in agreement with MODIS cloud top observations (Fig. A5). The reduced stability during 25 and 26 Nov 2014 points towards strong vertical mixing up to the mid-troposphere, potentially by surface heating, but due to dry convection only, as no clouds form, moisture is not depleted (Fig. 8).

### 4.2.2 Moisture sources

The moisture supply for precipitation during HPE5 originates dominantly from two major sources: one is to the south of the

study region, over the southern Sahara and Sahel, contributing to the first part of the event (HPE5.1; Fig 9a,c), and the other is the Atlantic Ocean representing the dominant moisture supply for the second part of the event (HPE5.2; Fig 9b,c). The MSD is only able to identify ∼70 % of the moisture origin for the first part of HPE5.1 (Fig 9c), pointing towards important contributions of moisture from uptakes that occurred more than ten days before the precipitation event. This is remarkable because typical water vapour residence times comprise durations of four to five days globally, with a trend to smaller duration in the tropics and

oceanic subtropics (Gimeno et al., 2021), while residence times of 8 days or more were found for precipitation in the Sahara (Sodemann, 2020). The boundary layer contribution is low with about 30 %, temporally averaged for the first part of HPE5.1, compared to the contribution of the free atmosphere. For the second part of HPE5.1, the MSD explains more than 90 %, and the boundary layer contribution rises up to about 40 %.

As the Saharan soils are very dry, the question arises, how come this region contributes so much moisture during the first part

of HPE5.1? We suggest here a moisture recycling mechanism we term as "domino effect", based on the MSD. Moisture from the tropics forms clouds and precipitates along the tropical plume. A large part of this precipitation evaporates or sublimates when falling into the relatively dry lower and mid-troposphere (possibly forming virgae). This moisture thus gets recycled and transported further north into the catchment in a domino-like process, which may repeat itself several times on the transport pathway from the tropics into the Sahara and finally into the catchment. This process may be reinforced by the convergence of

moisture from different tropical sources (tropical North Atlantic and from the Sahel, Fig. 8; similar observations as in Knippertz et al., 2003). Since no surface precipitation is detected by IMERG it is difficult to assess how important this domino effect induced by repetitive precipitation evaporation underway is. To confirm the significance of this process, in-situ observations, such as from radar observations, would be needed.

Nevertheless, two observations in ERA5 contribute to this hypothesis. First, the vertical time evolution of the atmosphere

above the catchment before HPE5.1 shows elevated values of specific humidity at mid-levels and cloud formation at higher levels already at 15 UTC on 20 Nov 2014 (Fig. 8b-c). It is assumed that these high-level clouds produce some precipitation





**Figure 7.** Overview of synoptic-scale conditions during HPE5.1 (panels a and b), and HPE5.2 (panels c and d). Total column water ($TCW$) is coloured, black thick contour is the 2-PVU line on 320 K isentrope, violet contour modelled precipitation (ERA5; where intensity is $> 0.5\,\mathrm{mm\,h^{-1}}$), orange depicts the location of the surface level cyclone, grey arrows are 850 hPa wind vectors (shown only where wind speed is $> 10\,\mathrm{m\,s^{-1}}$), grey contours are mean sea level pressure, red contour is the Sebkha el Melah catchment, brown shading shows the Atlas Mountain Range.

that gets totally evaporated and sublimated in the mid-tropospheric layer (400 - 800 hPa). This observation lets us assume that pre-moistening of the atmosphere is important for precipitation formation over Sebkha el Melah, and that recycling of moisture is possible in this region.

The second hint towards the existence of the domino recycling effect is based on the vertical moisture profile along the ERA5 trajectories crossing the region in the southwest of the catchment (Fig. 10c). It shows that on the way over the Sahara, a moist mid-level troposphere sporadically exhibits clouds and precipitation (Fig. 10). This signal is remarkably similar to the moisture recycling within the catchment as discussed above. Furthermore, it is in line with the MSD observation, showing that only a smaller part of the moisture uptake is attributed to evaporative sources in the boundary layer, the rest coming from



**Figure 8.** Temporal evolution of the atmospheric column during HPE5.1 and HPE5.2 (LFE5) in ERA5 interpolated to the catchment centre (Béchar; Fig. 1). (a) Time evolution of relative humidity ($RH$, turquoise line left axis) and precipitation ($P$, blue bars right axis). (b) Time evolution of specific humidity ($q$, coloured field), equivalent potential temperature ($THE$, black lines), 0°C-level (red line), boundary layer height ($BLH$, yellow line), and $10\,\mathrm{mg\,kg^{-1}}$ isolines of ice water content ($IWC$, solid light blue line), snow water content ($SWC$, dashed dark blue line), liquid water content ($LWC$, solid light grey line), and rain water content ($RWC$, dashed dark grey line). (c) Time evolution of $IWC$ (orange), $SWC$ (violet), $LWC$ (green), $RWC$ (blue), $THE$ (black lines), 0°C-level (red line), and $BLH$ (yellow line).

above the boundary layer possibly through dry convective mixing or, as hypothesised here, by below cloud evaporation and sublimation of hydrometeors falling out of the elevated tropical plume. Considering this process, the Sahara can be described as a moisture source induced by precipitation evaporation and not due to contributions from surface evaporation. This domino process should be further investigated in the future by numerical tracer experiments that allow to follow the moisture from surface evaporation in specific regions (e.g., Dahinden et al., 2023)

For HPE5.2, the MSD signal indicates an intense evaporation area over the North Atlantic as the dominant moisture source of precipitation (Fig. 9b,c), attributable to the intense extratropical cyclone approaching from southern Greenland (Fig. 7).



The largest precipitation amount arrives during the night from 28 to 29 Nov 2014 with $> 70\,\%$ of moisture coming from the North Atlantic, coinciding with the deep cyclone reaching the catchment from the west (Fig. 7). Furthermore, the boundary layer contribution varies between $55\,\%$ and $90\,\%$, which can be related to surface evaporation over the North Atlantic. As the deep cyclone on 28 to 29 Nov 2014 is moving rather slowly over the catchment, moist air parcels have sufficient time to move around the southern tip of the Atlas Mountains.

On 27 Nov 2014 though, a large portion of the precipitated moisture originates from just upstream of the catchment region. The building up of low-level clouds on 27 Nov 2014 (Fig. 8) hints towards the fact that the remaining moisture of HPE5.1 fuels the initial phase of precipitation during HPE5.2, further supported by an initial boundary layer contribution of up to $60\,\%$ to precipitation, supporting the hypothesis about the Saharan recycling process and the general importance of the remaining low-level moisture in the catchment. The MSD for HPE5.2 identifies $95\,\%$ - $100\,\%$ of the moisture sources (Fig. 9c).

To conclude this case study, it seems that a rather stationary extratropical low-level cyclone with an upper-level forcing positioned at the western coast of Morocco is essential to generate a strong enough HPE that triggers a LFE. This constellation, depending on its strength and exact position, may either induce moisture convergence (e.g., a tropical plume) from the tropics and the Atlantic into the catchment of Sebkha el Melah, or transport large amounts of oceanic moisture around and partly over the Atlas Mountains into the catchment. In both cases, pre-moistening of the atmosphere and recycling of the moisture within the Sahara seems to have a key impact on the precipitation amount. Additionally, the stationarity of the synoptic weather system, or the continuing reinforcement of a low-level extratropical cyclone allows moist air parcels to be advected and lifted during long enough time periods to create HPEs. The coupled or sequential lifting processes through upper-level forcing, orographic lifting, and potentially convection because of diurnal surface heating most probably play an important role in maintaining precipitation over a long enough time period to trigger a LFE.

### 4.3 Climatology of HPEs

Given the different precipitation forcing mechanisms that appear to play a role for just one exemplary LFE, as well as the observation of precipitation appearing dominantly during afternoon and night hours, in the following section we explore whether these are general characteristics of HPEs and LFEs in Sebkha el Melah.

LFE-generating HPEs have distinct characteristics compared to other HPEs and the mean climatology. To show these differences we divided the identified HPEs into three categories based on their HPE magnitude (Sect. 3.2.1): (a) LFE-generating HPEs, (b) strong HPEs, and (c) medium HPEs (Fig. 5a).

First, we examine the daily cycle of precipitation; it shows a clear diurnal pattern in all seasons (Fig. 11), albeit less pronounced in winter. Precipitation intensity rises in the afternoon, peaks in the late evening (18:00 to 21:00) and then drops back to lower values during the night and morning. This daily cycle indicates a major convective influence on total rainfall, due to the strong heating of the surface in the Sahara. During winter though, the precipitation intensity varies more throughout the day, suggesting that the convective influence is smaller, and the large-scale forcing from extratropical systems is more prominent.

Rain intensity during LFE-generating HPEs is higher compared to the intensity during strong and medium events, while exhibiting the same general daily cycle. In autumn, the difference in mean catchment intensity of LFE-generating HPEs compared




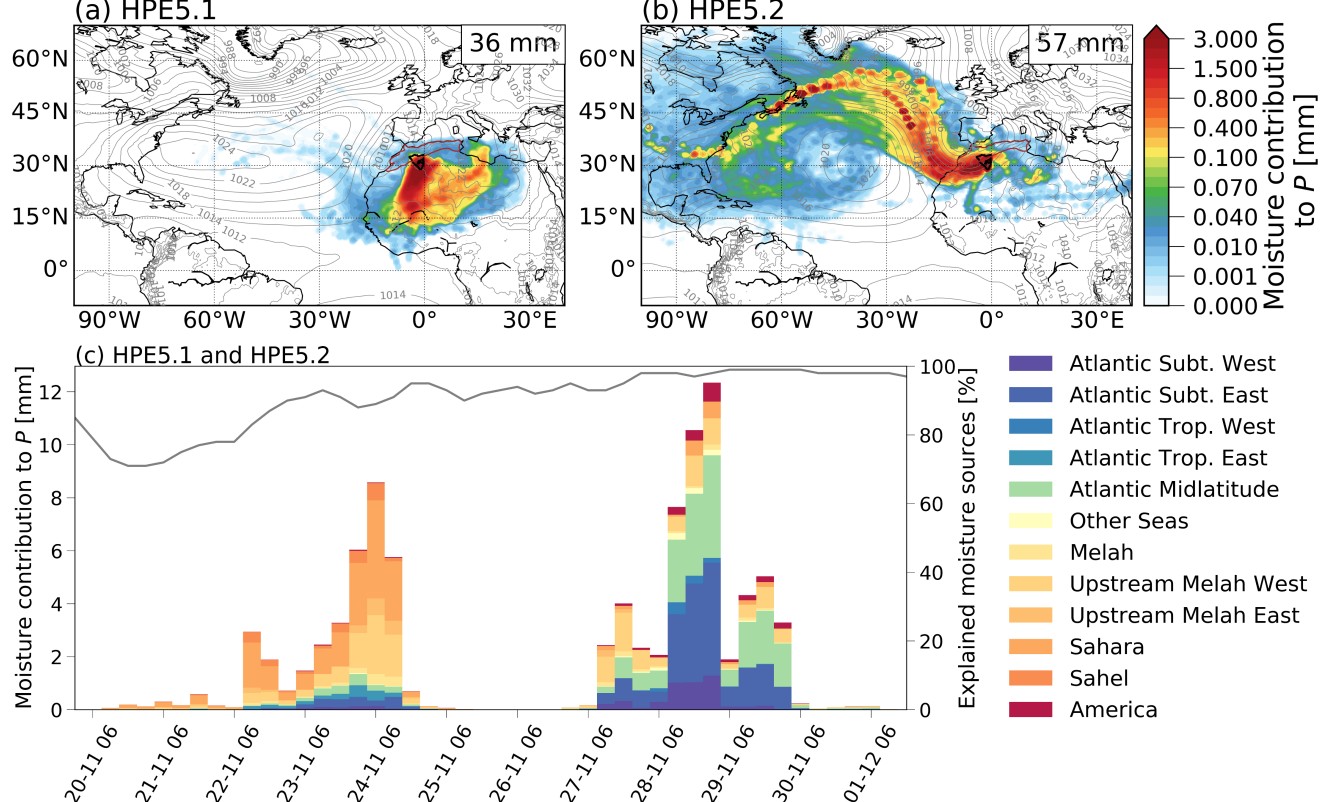

**Figure 9.** Moisture supply to precipitation during (a) HPE5.1 and (b) HPE5.2, and their (c) temporal evolution of regional partition (Fig. 4b), showing the major moisture sources (regions with moisture supply of $< 0.01\,\mathrm{mm}\,6\,\mathrm{h}^{-1}$ are excluded) and the percentage of explained moisture source (grey line). In panels a and b, a thick black contour indicates the Sebkha el Melah catchment, grey contours the mean sea level pressure isolines averaged over the HPE duration, and the upper-right box shows mean catchment precipitation. Generally, one pixel with 1 mm of moisture signal is interpreted as contributing to 1 mm of $P$ to the HPE over the catchment.

to other HPEs slightly diminishes, with LFEs situated roughly at the $65^{\mathrm{th}}$ quantile (Fig. 11). We hypothesise that this might be related to the typical area of rain cells, which is smaller in autumn, because of a larger contribution of convective precipitation caused by the smaller-scale forcing for precipitation; more isolated convective cells forced by surface heating show typically higher rain intensities, smaller areal coverage, shorter lifetimes, and smaller vertical velocities (e.g., Sharon, 1972; Belachsen

et al., 2017; Armon et al., 2018). Despite the small area covered by convective storms, those events may create LFEs if they are situated over the areas contributing efficiently to the lake-filling (i.e., the rockier and/or downstream part of the catchment). Yet, their precipitation intensity will be small when averaged over the catchment within the daily cycle.

In general, all three HPE categories exhibit the same type of synoptic-scale anomalies, but with an increased anomaly for higher HPE intensities (Fig. 12). The upper-level geopotential height anomaly during HPEs shows a major negative signal



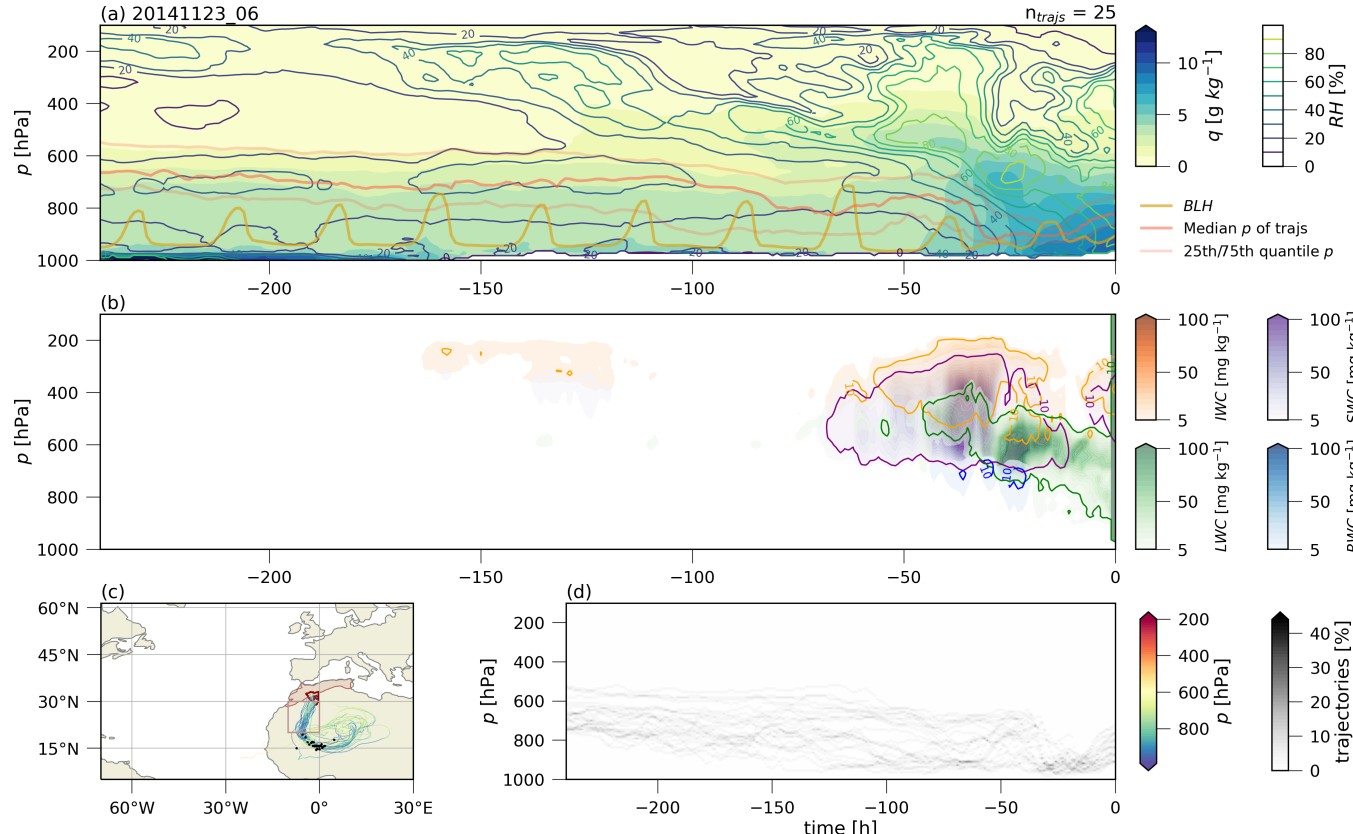

**Figure 10.** Vertical profile along the main moisture transport pathway at 06 UTC on 23 Nov 2014 (HPE5.1). (a) Vertical evolution of specific ($q$, coloured field) and relative ($RH$, coloured lines) humidity, the boundary layer height ($BLH$, yellow thick line), and averaged trajectory-pressure levels (red-shaded lines). (b) Vertical evolution of ice water content ($IWC$, orange), snow water content ($SWC$, violet), liquid water content ($LWC$, green), and rain water content ($RWC$, blue). (c) Trajectory positions and pressure levels (coloured) 10 days backwards, black dots indicate air parcel positions at t = -50 h, the red contour is the Sebkha el Melah catchment, and the brown contour is the Atlas Mountains. (d) Percentage of trajectories at every pressure level 10 days backwards.

centred to the northwest of the catchment over the western Atlas Mountains and the west coast of Morocco, accompanied by an upper-level cyclonic wind anomaly, and increased atmospheric moisture contents. These anomalies become larger for the strong HPEs and are the deepest for LFE-generating HPEs. The structure of these anomalies indicates the presence of an upper-level low or a trough, which are chiefly associated with $PV$ features such as streamers and cut-off lows (Fig. A6) and a more meandering jet stream (Portmann et al., 2021). These upper-level anomalies tend to destabilise the low-levels and

increase the potential of a surface cyclone formation, which is usually present at the west coast of Morocco (Fig. A7) during precipitation events in the catchment.



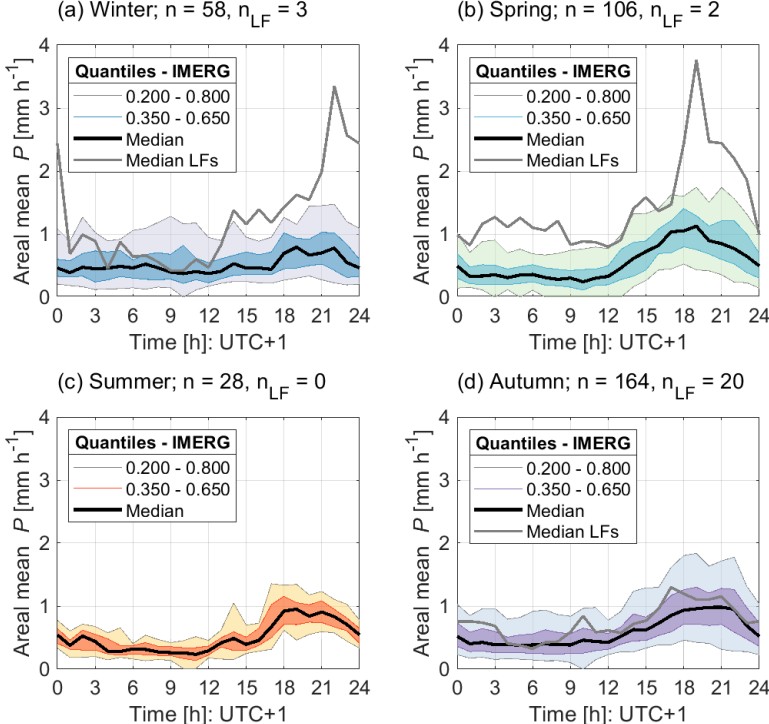

**Figure 11.** Daily cycle of mean catchment precipitation ($P$) intensity for HPE days for (a) winter (DJF), (b) spring (MAM), (c) summer (JJA), and (d) autumn (SON). Precipitation quantiles are computed compared to all HPE days. Grey lines show the median intensities for LFE-generating HPEs. n = number of HPE days. $n_{LF}$ = number of LFE-generating HPE days.

Upper-level wind anomaly at the western side of the catchment changes from west-south-west during medium HPEs (Fig. 12d), through southwest during strong HPEs (Fig. 12c) to south-south-west during LFE-generating HPEs. Similar patterns are exhibited at the surface as well, although with a lower intensity (Fig. A7). The normal westerlies (Fig. 12a) and the
southwesterlies during medium HPEs impinge on the Atlas Mountains, which may lead to increased precipitation on the windward side of the mountains and decreased precipitation on the lee-side (e.g., Roe, 2005; Marra et al., 2022). In contrast, during strong HPEs and LFEs, the stronger winds are blowing clearly around the Atlas Mountains, leading to decreased orographically-induced rain-out. Thus, larger amounts of moisture can reach the catchment and the southern side of the Atlas Mountains, enabling increased precipitation at the "rain shadow" of the mountains.
Atmospheric moisture content also shows clear anomaly signals during HPEs. Total column water increases over the catchment and upwind (southwest) of it, with larger anomaly amplitudes during strong and LFE-generating HPEs (Fig. 12). During LFE-generating HPEs, higher moisture values are seen near the Moroccan coastline, over the Atlantic Ocean, as well. This



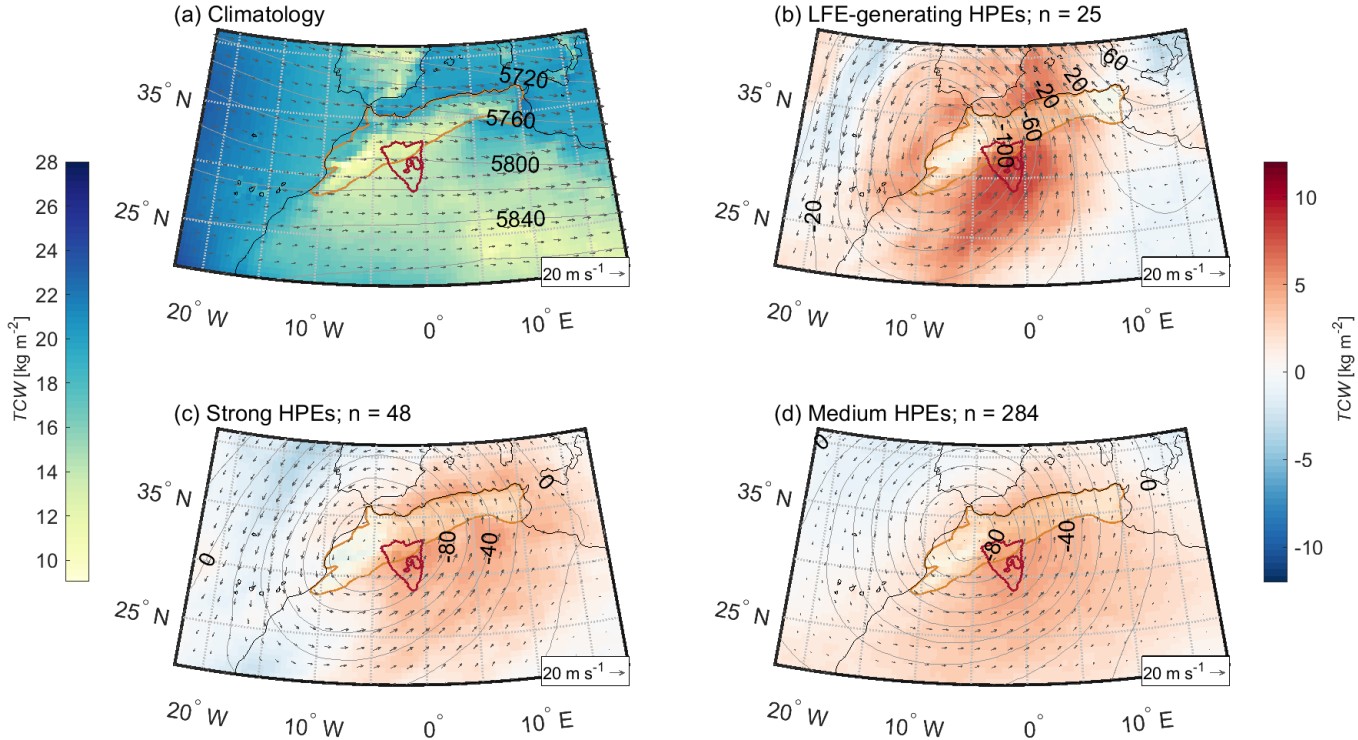

**Figure 12.** Upper-level and moisture anomalies during HPEs. (a) Average annual climatology of total column water ($TCW$; $\mathrm{kg\,m^{-2}}$; colours), winds at $500\,\mathrm{hPa}$ ($\mathrm{m\,s^{-1}}$; grey arrows), and geopotential height at $500\,\mathrm{hPa}$ (grey contours). Anomalies of these variables during LFE-generating HPEs (b), strong ($\geq 80^{\mathrm{th}}$ quantile) HPEs (c), and medium HPEs (d) are presented in the other panels. Red contour represents the Sebkha el Melah catchment, brown patch the Atlas Mountain Range. n=number of days considered.

emphasises the importance of moisture presence both over the Sahara (to the southwest of the catchment) and over the North Atlantic.

In addition to anomalous wind, pressure and humidity, HPEs are characterised by a negative surface temperature anomaly. Anomalously low surface temperatures ($T$ at $2\,\mathrm{m}$) are evident through all HPE classes, with a larger deviation for strong HPEs and even larger for LFE-generating HPEs when compared to the yearly climatology (Fig. A7). Compared to the wet-season (Sep-May) climatology (Fig. A8) the negative anomaly is accentuated over the southwest of the catchment. This negative anomaly may represent below-cloud evaporative cooling, consistent with the domino-moisture recycling process described in

Sect. 4.2, advection of cold air from the ocean, or decreased surface warming due to the presence of clouds. However, taken together, evaporative cooling and the positive moisture anomaly upwind and over the catchment suggest that the domino effect (Sect. 4.2.2) is likely essential in the generation of strong HPEs and LFEs.




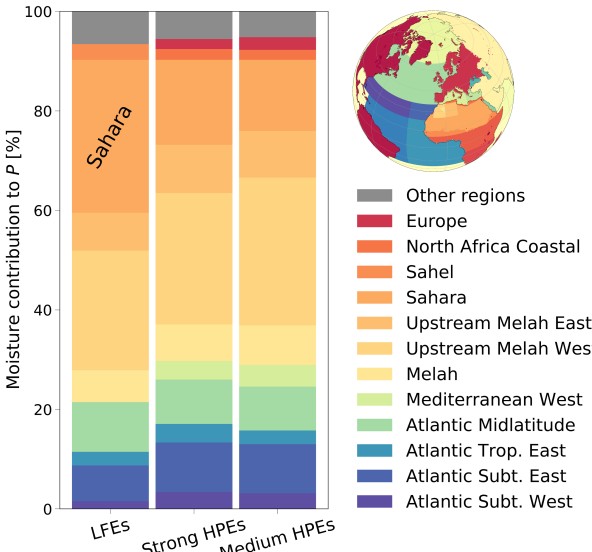

**Figure 13.** Composite climatology of moisture sources to HPEs. Columns represent contributions of different regions (legend and map on the right side, and in Fig. 4b) for LFE-generating HPEs (left), (b) strong HPEs, and (c) medium HPEs. Regions displayed are only those that contributed more than 1.5%. Values represent the attributed proportion of all precipitation during all HPEs in every category.

The importance of moisture recycling in this domino process seems considerable and intensifies with increased event magnitude. The origin of more than 60 % of the moisture throughout all HPEs is classified as coming from the desert (Fig. 13: orange-yellow colours). While the largest moisture-contributing region to HPEs is the Sahara and the vicinity of the Sebkha el Melah catchment, during LFE-generating HPEs the role of the farther Sahara and even the Sahel is clearly more important than for strong and medium HPEs. During these non-lake-filling HPEs, the moisture originates from regions closer to the catchment and higher contributions from the Atlantic and the Mediterranean are observed.

The MSD analysis reinforces the synoptic climatology observations (e.g., Fig. 12), which indicates that the domino effect is more important for LFEs. Smaller events receive moisture mainly from local moisture recycling over the nearby desert region, with supplemental moisture originating from the Atlantic and the Mediterranean regions. However, in order to generate enough precipitation to fill the lake, moisture recycled from farther away is necessary, brought from the tropics and recycled over the Sahel and the Sahara.

The second most important moisture source for HPEs is the Atlantic Ocean (Fig. 13: blue-green colours). Extratropical cyclones from the (North) Atlantic control the wind fields and bring moisture around the Atlas Mountains, especially during strong (non-LFE) HPEs (middle column in Fig. 13). The strong northerly winds over the North Atlantic (Fig. 12) advect dry midlatitude air over the relatively warmer subtropical ocean surface, thus increasing oceanic surface evaporation (Aemisegger and Papritz, 2018).





## 5 Summary and discussion

The purpose of this study is to better understand the meteorological ingredients needed for the occurrence of LFEs in the northwestern Sahara. Using remotely-sensed precipitation data we identified 250 HPEs over the catchment of Sebkha el Melah. We showed that the HPE magnitude is a good predictor for determining whether a HPE leads to the filling of the lake, with LFEs being induced by eight of the ten largest HPEs. In the 21 years between 06/2000 and 05/2021, these eight large HPEs resulted in six LFEs in Sebkha el Melah, with most LFEs resulting from only one HPE. These LFEs were quantified by applying

the MNDWI to the time series of Landsat imagery and comparing the inundated area with the hypsometric curve of the lake. Higher precipitation amounts were found to lead to higher runoff coefficients, albeit large inter-event variability exists. The meteorological conditions during HPEs were evaluated based on the ERA5 reanalysis data, first for an exemplary high-magnitude LFE that occurred in November 2014, and then for all of the identified HPEs, stratified by their magnitude: LFE-triggering HPEs, strong HPEs, and medium HPEs. We have assessed the atmospheric ingredients prevailing during precipitation, and

highlighted the most important factors related to HPEs in general and LFE-triggering HPEs in particular. In addition to the various atmospheric variables analysed, we computed backward trajectories and tracked the sources of moisture contributing to precipitation.

### 5.1 Ingredients for HPEs in the northwestern Sahara

Two meteorological features are responsible for generating heavy precipitation in the northwestern Sahara. These features

are normally absent, thus keeping the region generally dry. These features are i) synoptic-scale promotion and ii) meso-scale reinforcement for the *ascent of air*, in combination with *abundant moisture*.

**Ascent induced by synoptic- and mesoscale systems**. Processes at various scales are involved in the HPEs in the northwestern Sahara, ranging from the synoptic-scale to the local-scale. At the synoptic-scale, we have shown the importance of $PV$ structures and low-level extratropical cyclones (Sects. 4.2-4.3), which were previously also found to be important for different

parts of Morocco (Chaqdid et al., 2023). Upper-level $PV$ structures destabilise the lower atmosphere and give rise to ascent. Furthermore, both the upper-level $PV$ structures and low-level extratropical cyclones induce winds, which favour the advection of moist air from the Atlantic, the lifting of air parcels and the formation of deep clouds due to low-level convergence and the advection of moist air towards the topography of the Atlas Mountains.

Climatological analyses of upper-level $PV$ cutoffs and low-level extratropical cyclones show that the formation of such

systems at the Atlantic coast of Morocco is rather exceptional. The frequency of both upper-level $PV$ cutoffs and surface cyclones near the Atlantic coast of Morroco is roughly <5 % to 10 % with the highest frequencies during winter and autumn (Wernli and Schwierz, 2006; Portmann et al., 2021). However, these two features do not necessarily co-occur, which highlights the exceptional situation needed for the formation of HPEs over the northwestern Sahara in today's climate. This finding may help to investigate past and future projections of HPEs over the southwestern Sahara in particular in relation to shifts in the

jet stream, Rossby Wave breaking leading to the occurrence of upper-level $PV$ cutoffs and surface cyclones (e.g., de Vries, 2021).





On smaller scales, lifting mechanisms are connected to the wind field induced by the cyclone at the western coast of Morocco. Southerly to southwesterly winds force air parcels to travel over the Atlas Mountain chain. This either directly lifts air parcels enough to generate orographic precipitation or helps to overcome the convective inhibition that suppresses free convection. Furthermore, wind convergence can reinforce this ascent of air parcels. When convergence occurs upwind of the catchment (e.g., Fig. 7) it can support the formation of precipitation, inducing the moisture recycling process described above as a domino effect.

On the smallest scale, the daily cycle of HPEs (Fig. 11) indicates a contribution of free convection to the precipitation formation with peak intensities observed in the evening in all seasons. However, it must be noted that convective precipitation is probably not sufficient to generate significant HPEs and LFEs; no LFEs are identified when the surface is warmest, i.e., in summer, and in the other seasons convection happens mainly in connection with a synoptic-scale system (e.g., Fig. 12). Therefore, while short-lived deep convection is important for HPEs, triggering and enhancing precipitation in the northwestern Sahara, it requires larger scale forcing as well.

**Moisture abundance**. The most important moisture sources for precipitation in the region are: (1) local sources from precipitation recycling within the catchment and its surroundings ($\sim 65\,\%$), (2) remote sources such as the North Atlantic, the Sahel, or the Mediterranean ($\sim 30\,\%$ combined; Fig. 13).

In the desert, the importance of nearby moisture sources may seem surprising as actual evaporation from the surface is close to zero. However, we suggest that this moisture is actually mostly recycled; moisture diagnosed here as being taken up over the Sahara does not come from surface evaporation, but from moistening through precipitation evaporation and sublimation, and mixing due to convergence of air parcels and convection (Dahinden et al., 2023). This means that this moisture might have originally come from other sources compared to the ones identified here, most likely from the tropics and the North Atlantic – upwind of the identified moisture sources. Cloud formation in squall lines or other mesoscale convective systems and tropical plumes are potential candidates which fuel moisture transport into the southwestern outskirts of the Sahara, where it can precipitate, be recycled, and be transported deeper into the desert by the southerly flow. This sequence is repeated in a domino-like manner until the catchment region is reached, thus contributing 'Saharan-flagged' moisture to the HPEs in our applied moisture source diagnostic approach. In addition, this aerial precipitation recycling process is supplemented by continental recycling, thus surface evaporation of the moisture remaining from recent precipitation events, which is common mainly in autumn (see e.g., the November 2014 case study [Sect. 4.2] and Fig. 8, and Fig. A3).

While several types of synoptic or mesoscale systems can contribute to cloud formation upwind of the Sahara, to transport this moisture deep into the desert, persistent southwesterly flows are needed. These flows occur in the mid- to upper-levels of the troposphere whenever a southward intruding upper-level trough is present at the edge of the northwestern Sahara or over the Atlantic coast, which is indeed the case during the identified HPEs (Fig. 12). The related cyclones accumulate moisture from the Atlantic Ocean over their life cycle and precipitate during their passage into the Sahara and the catchment of Sebkha el Melah. This type of tropical–extratropical interaction was previously associated with tropical plumes (Knippertz et al., 2003; Knippertz, 2003; Rubin et al., 2007; Skinner and Poulsen, 2016).





## 5.2 Characteristics that differentiate between LFE-triggering, strong, and medium HPEs

Three conditions seemingly determine LFEs. These are the (a) lifting mechanism, (b) wind conditions, and (c) moisture source, which are described below.

LFEs need a strong and long-lasting lifting mechanism, which is maintained by the coupling of large-scale forcing and the
mesoscale convective forcing. An example of this is given by the November 2014 case study (Sect. 4.2) where $PV$ structures appear repeatedly in the vicinity of the catchment. The persistent conditions allow for increased moisture transport, continuous formation of rain cells, and therefore also for a longer rainfall duration and areal coverage, which are needed to generate intense desert floods (e.g., Doswell et al., 1996; Morin and Yakir, 2014). These conditions lead to relatively long-duration HPEs, typically 3 d for the LFE-triggering HPEs.

The wind conditions are related to the large-scale forcing. Orographic lifting only appears when the wind is strong enough and directed close to perpendicular towards the Atlas Mountains (i.e. southerly winds, similar to what was shown by Froidevaux and Martius (2016) for heavy precipitation in the Alps), such that the air parcels are forced to rise, making it possible to overcome the typical subsidence aloft. However, it seems that the essential ingredient is not the south-to-southwesterly wind within the catchment but actually, the southwesterly flow that circumvents most of the High Atlas from the south such that the
moisture is not being lost to precipitation beforehand. These winds blow with higher velocities when the low-level cyclone is relatively deep and situated close to the west coast of Morocco (Fig. A7).

Lastly, to obtain enough moisture for creating heavy precipitation, it needs to be transported not only from the surrounding regions but also from far away – tropical Africa and the midlatitude North Atlantic – being transported and recycled by the domino effect once it has reached the edge of the Sahara. Favourable conditions for such long-range transport events are
also obtained when the winds are stronger and blow from the southwest. The non-lake-filling HPEs show a less pronounced recycling process of moisture in the Sahara. The major part of the moisture of these events originates from the near upstream region and the moisture uptake is in general weaker than for LFEs (Figs. 11, 13).

When all the above conditions are met, the precipitation that falls in the catchment of Sebkha el Melah has a higher potential to fill the lake. Namely, events are generally longer (2-5 days, compared with 1 day on average for the non-lake-filling HPEs)
and have high precipitation volumes. While the volume itself may not be sufficient to determine whether the lake gets filled, as few of the heaviest precipitation events did not fill the lake, the HPE magnitude (Sect. 3.2.1) is quite a good determinant for lake-filling. Out of the ten highest flood magnitude HPEs eight produced LFEs. This means that high-intensity precipitation needs to be rather spatially focused and located over the runoff-contributing areas of the catchment (Sect. 2) to overcome infiltration rates and generate high-magnitude surface runoff and flows in the streams.

## 5.3 From precipitation to runoff: effective runoff coefficients

Effective runoff coefficient, namely, the portion of precipitation that ends up in the lake, is one of the most important but non-resolved factors in relating paleo-lakes to paleo-climatic conditions (e.g., Quade et al., 2018). Runoff coefficients are generally not available for large desert catchments, especially at the event-scale (Merz et al., 2006; Enzel et al., 2015). At the




annual-scale, considering that it rains every year but the lake remains empty most of the time, effective runoff coefficients are
expected to be close to zero. The runoff coefficients presented here are, to the best of our knowledge, the first to be obtained
over the Sahara which represent the actual ratio between event precipitation and lake volume. Annual runoff coefficients in
AHP literature range from $\sim 10\,\%$ to $\sim 30\,\%$ (annual values; e.g., Kutzbach, 1980; Coe and Harrison, 2002; Enzel et al.,
2015; Bouchez et al., 2019). Here we show that the event-based effective runoff coefficients in the basin range over about 5
orders of magnitude, from $\sim 1 \times 10^{-4}\,\%$ to $\sim 1 \times 10^{1}\,\%$ (Fig. 5b). These values are in line with the results of Wasko and Guo
(2022) based on gridded rainfall data in Australia, but only a few of the catchments in their study are large, arid catchments.
The construction of dams in the catchment is expected to lower the effective runoff coefficient. However, large HPEs are not
expected to be heavily affected; the main dam in the catchment can only contain $\sim 0.25\,\mathrm{km}^3$ (Sarra et al., 2023), which is $\sim 4\,\%$
of the mean catchment rainfall in the largest HPE (Fig. 5a).

Desert lakes are commonly relatively small compared to their catchment area, therefore, effective runoff coefficients control
the filling of these lakes, with only a minor filling contribution from direct rainfall over the lake (Quade et al., 2018). Therefore,
when reconstructing the water budget of a desert lake, slight shifts in the values of the effective runoff coefficient lead to large
deviations in the estimation of lake volumes. With the 21 years of data, the largest effective runoff coefficient values we
estimate here per event are on the order of $10\,\%$ and most are well below (Fig. 5b). These are very low values compared to the
ones used in estimating paleo-rainfall during the AHP based on hydrological and radiative budgets (e.g., Kutzbach, 1980; Coe
and Harrison, 2002). Here we showed that higher runoff coefficients seem related to higher event-rainfall depths and shorter
inter-event dry periods. Therefore, based on present-day observations, we suggest that the filling and maintaining of Saharan
lakes can be related to the intensity of individual events and changes in their frequency. I.e., if high-magnitude events are
frequent within a year, the annual runoff coefficient is expected to increase even if the annual rainfall remains the same. Since
it could take years for lakes to empty out through evaporation (e.g., Fig. 3c), a higher frequency of years with such frequent
high-magnitude events can maintain filled lakes with no change in the mean annual rainfall. We do not mean to imply that no
change in precipitation occurred during the AHP, but rather that the magnitude of the change might have been less important
compared to changes in the intensity and frequency of HPEs. One major implication of this suggestion is that to effectively
model precipitation changes during past (or future) periods in the Sahara, relatively small-scale changes in precipitation need
to be accounted for, calling for high-resolution model simulations. Given that precipitation during Saharan HPEs is largely
convective, as discussed in Sect. 4.3, in order to accurately simulate changes in both precipitation and runoff, models need to
be able to resolve convective-scale processes, which reaffirms conclusions made by both Folwell et al. (2022) and Jungandreas
et al. (2023).

## 6 Concluding remarks

Based on remote sensing of HPEs and LFEs, we show in this study the key ingredients involved in the most extreme HPEs that
lead to the filling of Sebkha el Melah in the northwestern Sahara (Fig. 14). These ingredients are:



- Anomalously deep, long-lasting- or a succession of extratropical low-level cyclones (Fig. 14, i) accompanied by an upper-level $PV$ feature (ii) to the west of the catchment. These enable both large-scale air ascent and the flow of air around the Atlas Mountains (v).

- Moisture convergence from the tropics (tropical plume) and North Atlantic (iii), and moisture recycling ("domino-process") upstream of the catchment (iv).

- Coupled or sequential lifting processes: orographic lifting (mountains), upper-level forcing, and surface heating (vi).

The combination of these ingredients not only enables the rare formation of rainfall in this arid region, but also helps in promoting heavy precipitation, floods, and the filling of Sebkha el Melah. Specifically, the most intense HPEs, the ones leading to LFEs, differ from less extreme HPEs by exhibiting stronger moisture convergence and advection of moist air around the Atlas Mountains. The recycling-domino-process plays a significant role in contributing high amounts of moisture to the catchment, enabling high precipitation volume. A small, but highly variable fraction ($\sim 1 \times 10^{-4}\,\% - \sim 1 \times 10^{1}\,\%$) of the precipitation volume makes it all the way to Sebkha el Melah. There, even without additional HPEs, water can persist for many months up to years. Other Saharan lakes, suggested to be filled in wetter periods in the past, are potentially contributed from similar processes; their initial filling and persistence might reflect a change in the intensity or frequency of events, rather than a change (only) in the mean precipitation.

In a nutshell, this study has shed light on the relevant combination of dynamical ingredients needed to form heavy precipitation in the northwestern Sahara that induces subsequent lake-filling. This knowledge is relevant for long-term water resource management strategies in Saharan countries. In addition, the insights we presented provide new fundamental atmospheric process understanding that can help in evaluating potential future and past greening of the Sahara.

*Code and data availability.* ERA5 data are available to download through https://cds.climate.copernicus.eu/cdsapp!/dataset/reanalysis-era5-single-levels?tab=overview. MODIS data used here is courtesy of the NASA Worldview application (https://worldview.earthdata.nasa.gov/), part of the NASA Earth Observing System Data and Information System (EOSDIS). Landsat data, courtesy of the U.S. Geological Survey, were obtained through Google Earth Engine, and are available via https://developers.google.com/earth-engine/datasets/catalog/landsat. IMERG data are available to download through the NASA Goddard Earth Sciences (GES) Data and Information Services Center (DISC). The bathymetric information of Sebkha el Melah was obtained from the results of Armon et al. (2020), and is available upon reasonable request from the corresponding author. An overview file of the properties of precipitation during all identified HPEs is available as supplemental material.

*Author contributions.* This paper is an outcome of JCR's MSc thesis supervised by MA and FA. MA designed the study. ED contributed Google Earth Engine codes, as well as support with Landsat data. JCR carried out most of the analyses based on codes by FA and MA, and wrote the original draft of the paper with feedback from FA and MA.



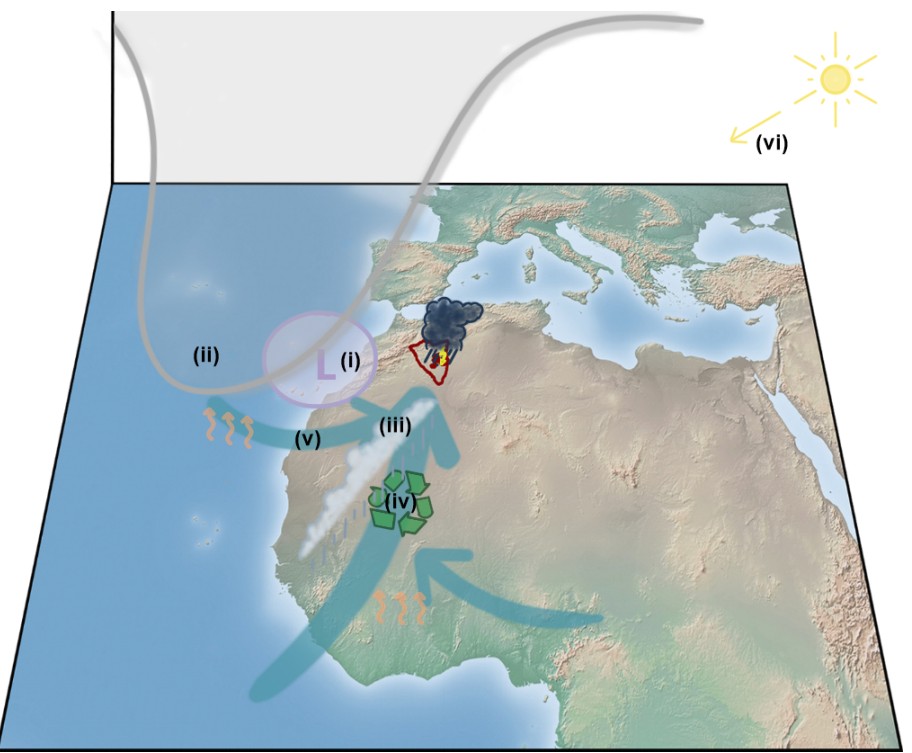

**Figure 14.** Schematic summary of the major ingredients involved in filling Sebkha el Melah: (i) extratropical low-level cyclone (violet patch), (ii) southward intruding upper-level $PV$ structure (grey), (iii) moisture convergence (turquoise arrows), (iv) moisture recycling over the Sahara (domino process), (v) southwesterly wind bypassing the Atlas Mountains and lifted orographically in the "rain shadow" side of the mountains, (vi) surface heating and convection triggering. The Sebkha el Melah catchment is marked in red. The basemap is made with Natural Earth.

*Competing interests.* The authors declare that they have no conflict of interest.

*Acknowledgements.* The authors would like to thank Heini Wernli for supporting this study and providing helpful feedback, as well as for partially funding it. We also thank Yehouda Enzel for helping us acquire runoff coefficient data. Special thanks go to Michael Sprenger for the very reliable technical support and for making the ERA5 dataset available for us, and to Lisa Gross for technical and design support in creating

Fig. 14. MA was supported by an ETH Zurich Postdoctoral Fellowship (Project No. 21-1 FEL-67), by the Stiftung für naturwissenschaftliche und technische Forschung and the ETH Zurich Foundation.





**Table 2.** Overview of LFEs.

| LFE | HPE date | Lake area identification | | Mean catchment $P$ | | Lake volume change estimate, MNDWI 0.4 (Min. MNDWI 0.7; max. MNDWI 0.1) | Effective runoff-coefficient[a] |
|---|---|---|---|---|---|---|---|
| | | before HPE | after HPE | $P_{\text{IMERG}}$ | $P_{\text{ERA5}}$ | | |
| LFE 1 | 26. to 27. Oct 2006 | 25. Oct 2006 | 12. Dec 2006 | 20 mm | 14 mm | $5.7 \times 10^{-2}$ km$^3$ ($4.2 \times 10^{-2}$ km$^3$; $7.0 \times 10^{-2}$ km$^3$) | 3 % |
| LFE 2[b] | 08. to 10. Oct 2008; 12. to 15. Oct 2008 | 08. Oct 2008 | 28. Nov 2008 | 89 mm | 49 mm | $7.6 \times 10^{-1}$ km$^3$ ($5.5 \times 10^{-1}$ km$^3$; $8.3 \times 10^{-1}$ km$^3$) | 9 % |
| LFE3[c,d] | 27. to 28. Mar 2009 | 04. Mar 2009 | 07. May 2009 | 49 mm | 23 mm | $-3.1 \times 10^{-2}$ km$^3$ ($-1.3 \times 10^{-1}$ km$^3$; 0 km$^3$) | <1 % |
| LFE4[c] | 18. to 20. Oct 2012 | 07. May 2012 | 01. Dec 2012 | 37 mm | 26 mm | $6.6 \times 10^{-5}$ km$^3$ ($4.6 \times 10^{-5}$ km$^3$; $3.2 \times 10^{-4}$ km$^3$) | 0.002 % |
| LFE5[b] | 21. to 24. Nov 2014; 27. to 30. Nov 2014 | 21. Nov 2014 | 16. Jan 2015 | 93 mm | 72 mm | $3.2 \times 10^{-1}$ km$^3$ ($5.6 \times 10^{-2}$ km$^3$; $3.6 \times 10^{-1}$ km$^3$) | 4 % |
| LFE6[c] | 16. to 18. Dec 2016 | 01. Oct 2016 | 28. Dec 2016 | 32 mm | 25 mm | $3.9 \times 10^{-4}$ km$^3$ ($5.2 \times 10^{-4}$ km$^3$; 0 km$^3$) | 0.01 % |

[a]Based on the IMERG event precipitation compared to the lake volume estimate of MNDWI threshold 0.4.
[b]Two HPEs are associated with this LFE.
[c]LFEs in which Sebkha el Melah was already inundated before the event.
[d]Detected via MODIS observations but not with the MNDWI identification (Sect. 3.2.2)





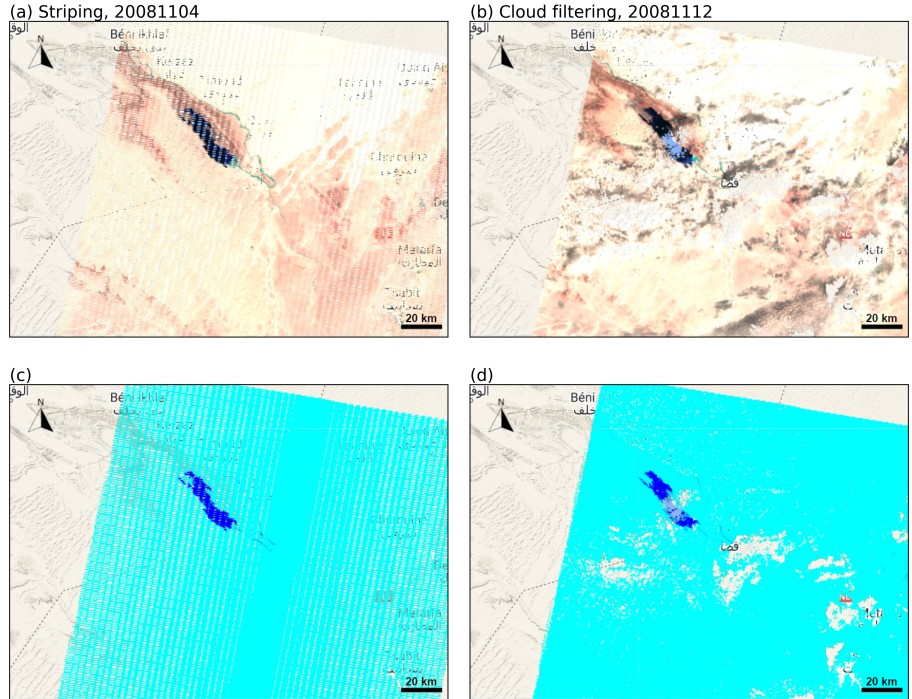

**Figure A1.** Examples of MNDWI-based lake area identification errors. (a) and (c) Landsat-7's striping errors, (b) and (d) cloud filtering errors. The blue layer represents the MNDWI-based water identification with dark blue indicating water areas.

**Appendix A**

**A1  Supplements**





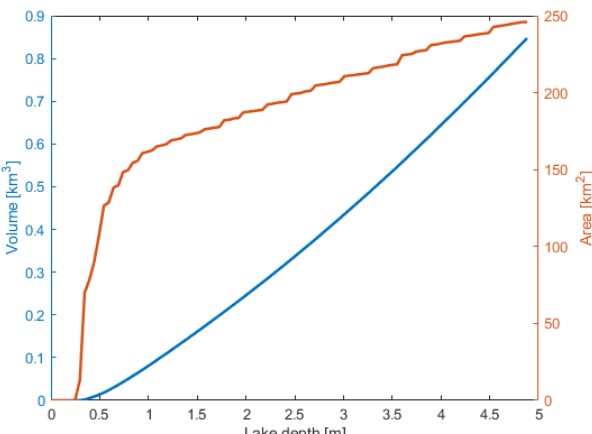

**Figure A2.** The hypsometric curve of Sebkha el Melah for lake area to volume conversion, based on the bathymetry estimates from Armon et al. (2020).

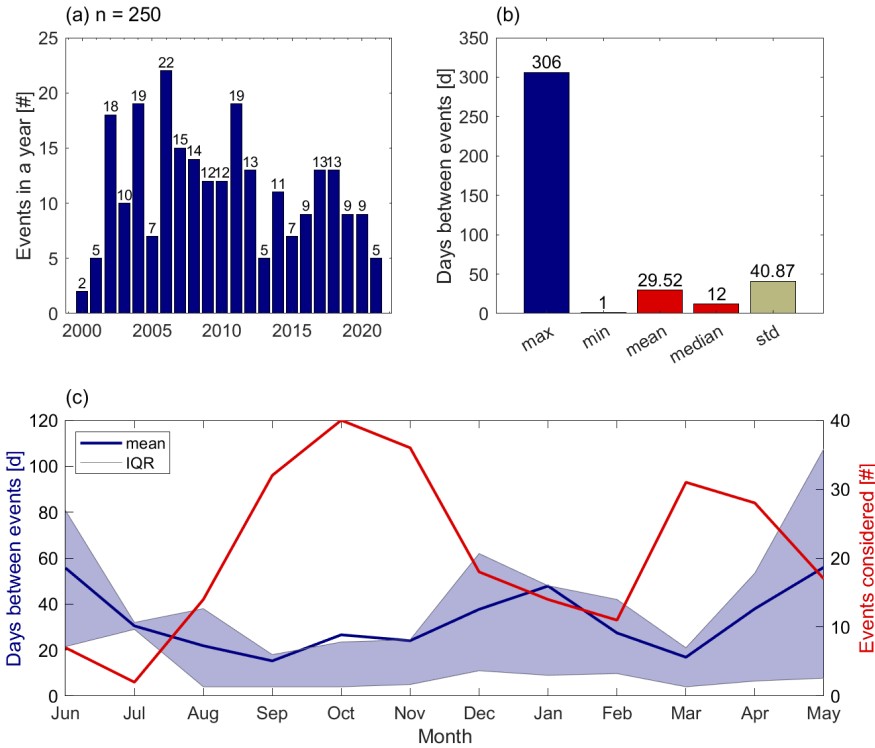

**Figure A3.** HPE occurrence (06/2000 – 05/2021). (a) Yearly number of HPEs (labelled bars). n=250 is the number of HPEs considered. (b) Statistical properties of the inter event periods (bars). (c) Yearly cycle of the inter event period (left axis; blue line=mean, blue shade=inter-quantile range [IQR]) and the identified HPEs (right axis; red line) for the observation period.

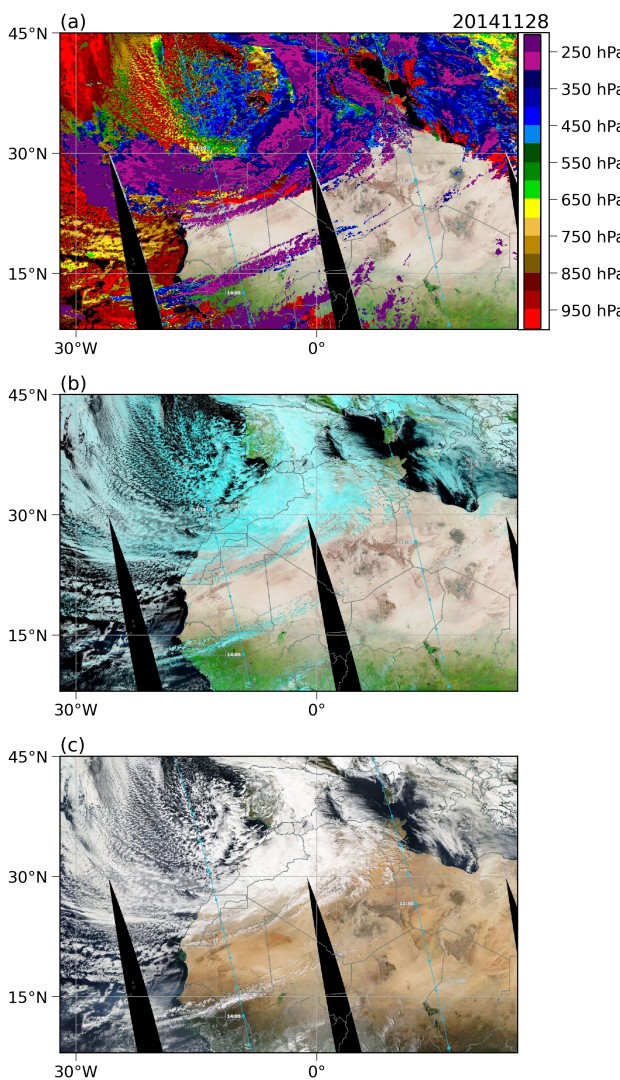

**Figure A4.** Cloud cover (28 Nov 2014; HPE5.2, Sect. 4.2) from MODIS Aqua satellite observations (NASA Worldview). (a) Cloud top pressure (CTP), (b) corrected reflectance bands 7-2-1, and (c) corrected reflectance true colour. The blue line indicates the satellite path and its crossing time.



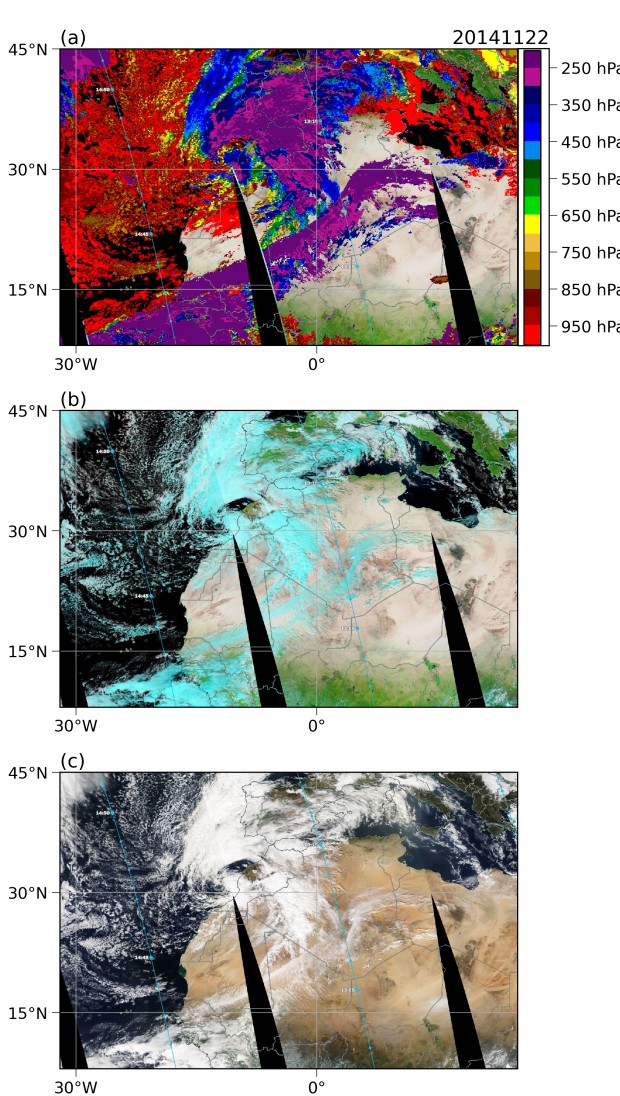

**Figure A5.** As in A4 but for HPE5.1 (Sect. 4.2), 22 Nov 2014.



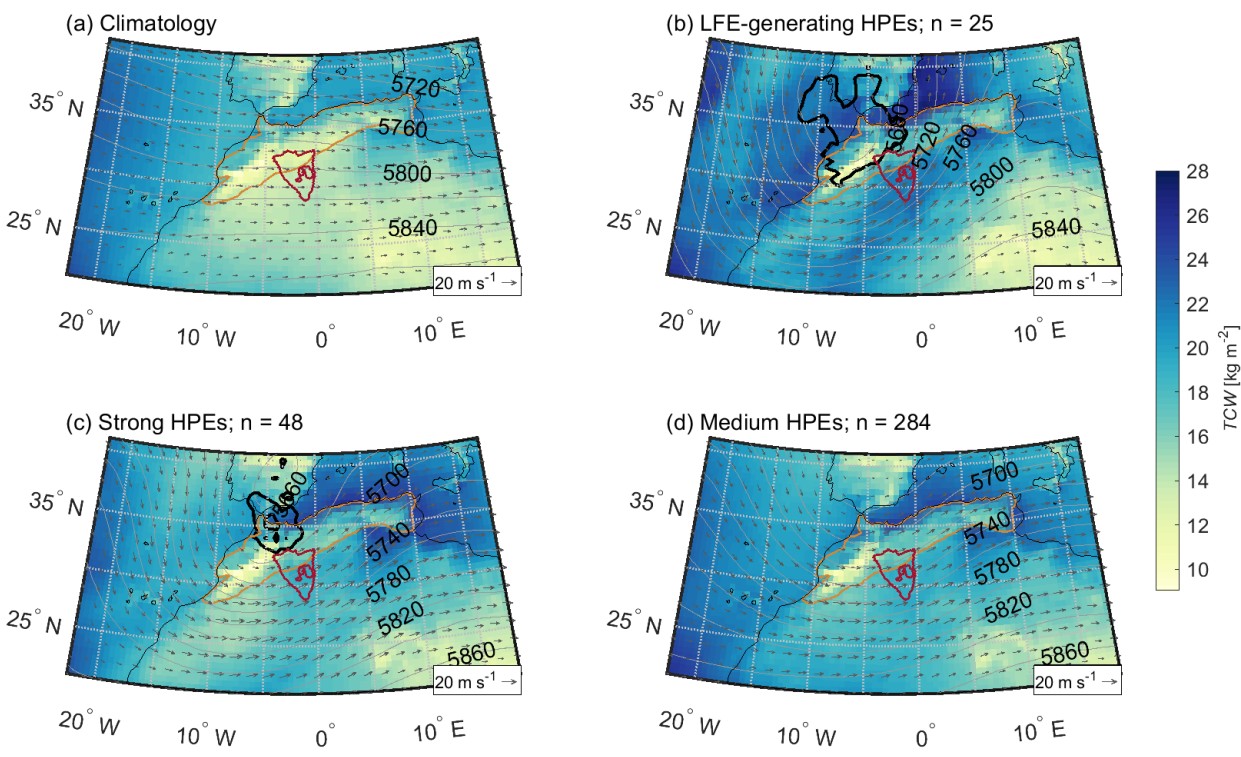

**Figure A6.** Climatology of total column water ($TCW$) ($\mathrm{kg\,m^{-2}}$; colours), winds at $500\,\mathrm{hPa}$ ($\mathrm{m\,s^{-1}}$; grey arrows), geopotential height at $500\,\mathrm{hPa}$ (grey contours), and the 2-PVU-line on $320\,\mathrm{K}$ (black contour) for (a) the period between June 2000 and May 2021, (b) the LFE-generating HPEs, (c) the strong HPEs, and (d) the medium HPEs. Red contour represents the Sebkha el Melah catchment, brown patch the Atlas Mountain Range. n=number of days considered.





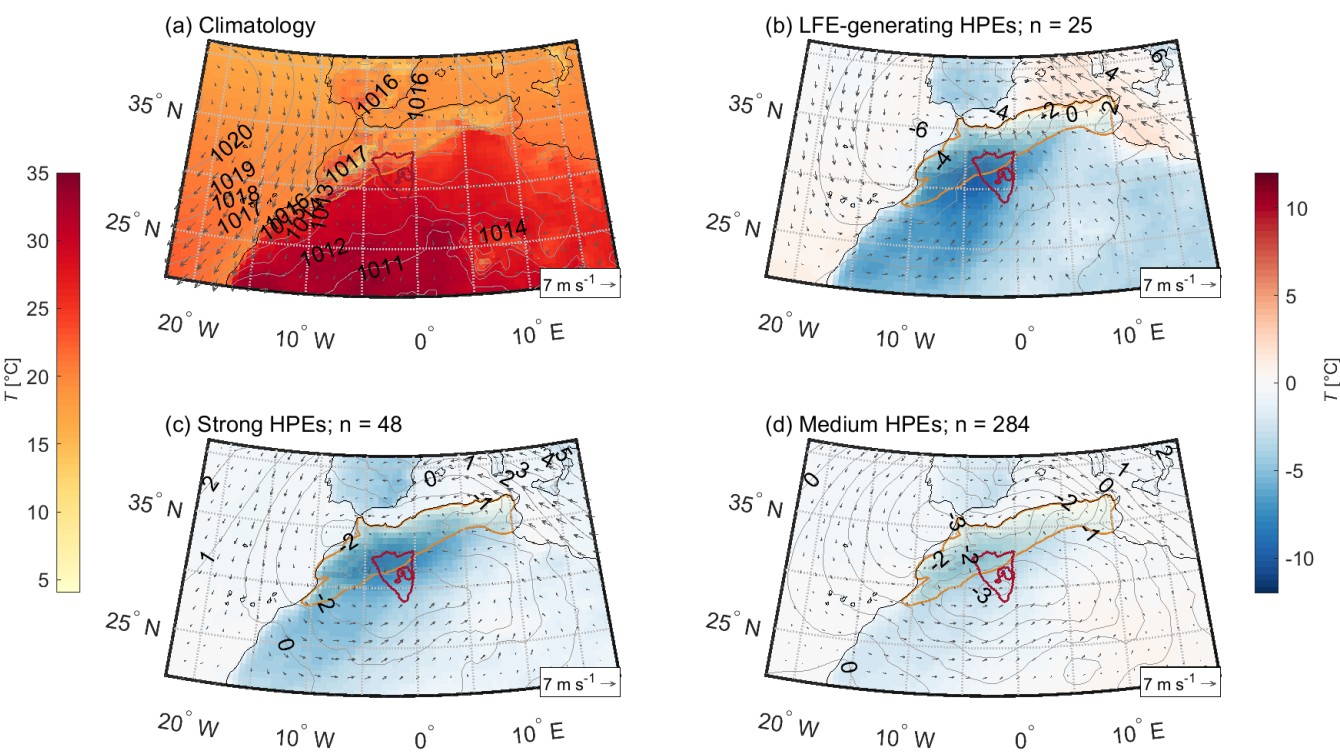

**Figure A7.** Climatology (a) and anomalies (b-d) of two-meter temperature ($T$) (°C; colours), winds at 10 m (m s$^{-1}$; grey arrows), and mean sea level pressure (grey contours) for (a) the period between June 2000 and May 2021, (b) the LFE-generating HPEs, (c) the strong HPEs, and (d) the medium HPEs. Red contour represents the Sebkha el Melah catchment, brown patch the Atlas Mountain Range. n=number of days considered.




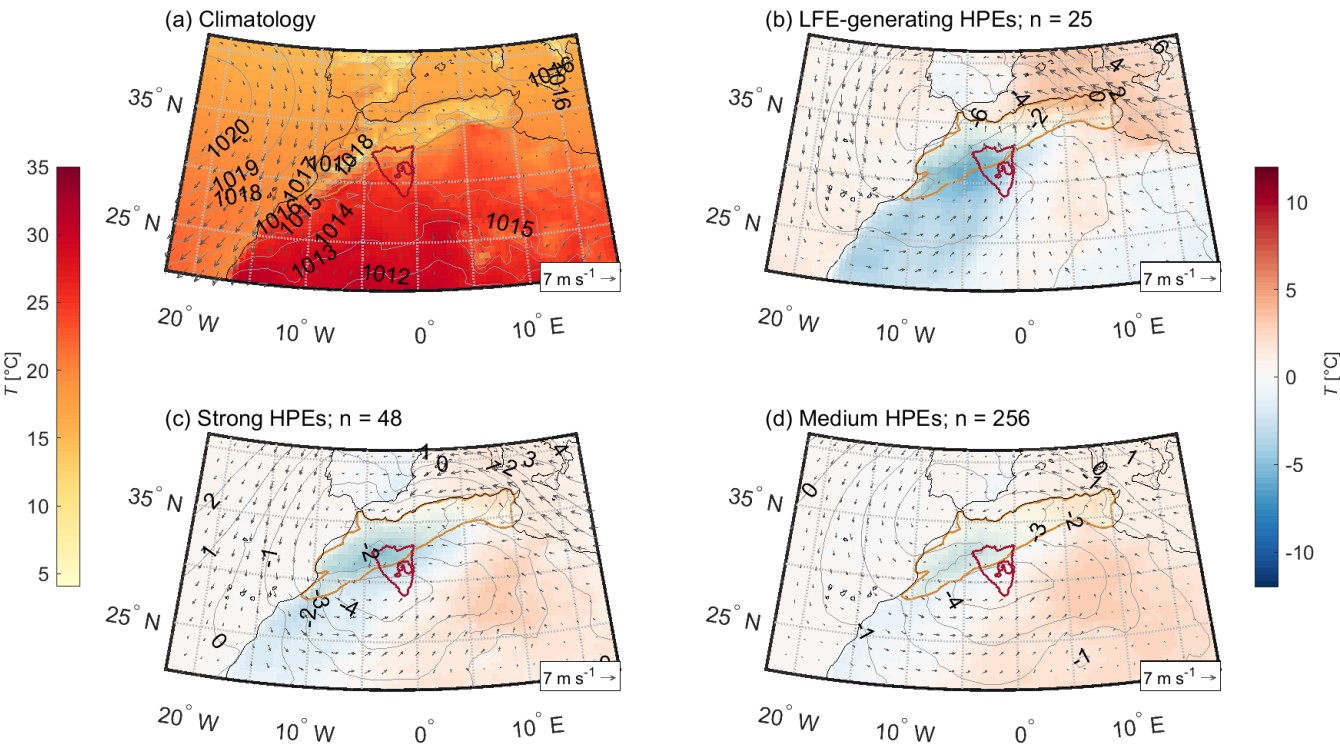

**Figure A8.** September to May climatology (a) and anomalies (b-d) of two-meter temperature ($T$) (°C; colours), winds at $10\,\text{m}$ ($\text{m s}^{-1}$; grey arrows), and mean sea level pressure (grey contours) for (a) the period between June 2000 and May 2021, (b) the LFE-generating HPEs, (c) the strong HPEs, and (d) the medium HPEs. Red contour represents the Sebkha el Melah catchment, brown patch the Atlas Mountain Range. n=number of days considered.





**Table A1.** Uncertainties in the assessment of lake area and volume.

| Uncertainty type | Explanation | Impact and mitigation | References |
|---|---|---|---|
| Area identification – MNDWI threshold | Depending on the MNDWI threshold chosen to identify wet pixels the lake area can vary | Small uncertainty expected, as the lake area identification shows small variations between thresholds of 0.25 to 0.55 | Fig. 3a |
| Area identification – Landsat striping | Void stripes in the satellite images occurs due to problems with Landsat-7, leading to missing data over the lake | Event area data points were selected manually to exclude damaged data, using either Landsat-5 or 8, except for LFEs #4 and #6 where area estimates are underestimated by up to 20 %, which translates into volume uncertainty smaller than evaporation / MNDWI thresholding uncertainties. | USGS Landsat Missions, Fig. A1 |
| Area identification – cloud filtering | Clouds can make shadows, leading to mis-interpretations of water surfaces. Thus, clouds are filtered out. When clouds are filtered over the lake, this area will be missing. | Small uncertainty expected, as this issue occurs only rarely and can be excluded with our manual event area data point selection | Pekel et al. (2016), Fig. A1 |
| Area identification – wind effects | Wind can push the water to be piled up over one side of the lake. This could lead to a reduction of the lake area. | Very small uncertainty is expected, especially when the lake depth is large, since a steeper topography prevents water from piling up. | Armon et al. (2020) |
| Volume identification – lake level | Lake morphology shows flat slopes at the bottom but steeper slopes at higher levels of the lake. This leads to higher sensitivity of volume estimates with changes in area at higher lake levels. | Higher uncertainties expected with larger lake area/volume | Armon et al. (2020), Fig. A2. |
| Volume identification – bathymetry | Lake bathymetry has on average a ~0.3 m RMSD error. | High uncertainties are expected for smaller LFEs, as their depths are relatively low (in some case less than 1 m), while low uncertainties are expected for the larger LFEs (depths of ~4.5 m). | Armon et al. (2020), Fig. A2. |



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
