# Peer review of "Meteorological ingredients of heavy precipitation and subsequent lake filling episodes in the northwestern Sahara"

_EGUsphere, 2024_

## Author Comment (AC1)

Reviewer #1

**Overview:**

The paper quantifies the meteorological and synoptic conditions during the filling of a lake in the northwestern Sahara using reanalysis data and satellite images. The paper describes the synoptic conditions leading to the lake filling and shows that cyclones and Sahara moisture recycling are important prerequisites for a lake filling event. They propose a shift in the understanding of the mid-Holocene green Sahara phenomenon, emphasizing the frequency and intensity of events rather than a large-scale northward shift of the rain belts.

I found the paper very interesting and innovative, and one that will have an important impact on the thinking of the greening of the Sahara. I think the paper is suitable for publication once a few minor, mostly structural and clarification comments, are addressed.

We would like to thank Reviewer #1 for the time taken to read our manuscript and for the constructive comments. We are also glad to see that we could convey the main messages of this manuscript in an understandable way and that they are acknowledged by the reviewer. Please see below our detailed response to all the comments raised by the reviewer.

**Minor comments:**

I found the structure of the results somewhat confusing. The paper would benefit greatly, if there is a better separation between the observations and interpretations and a clearer explanation of what the main points of the figures are. For example, in fig 7, instead of "Overview of synoptic-scale conditions during HPE5.1" use "An example of the formation of a southward moving deep cyclone during an HPE event". Or fig 5, instead of "Overview of.." use: "Duration, intensity and runoff coefficients of HPE and LFE events". And perhaps add commentary to what the reader should understand from the caption "showing an agreement of intensity and duration between ERA5 and IMERG data" etc. This is true for all figure captions. Please help the reader understand what the main point of the figure is.

We thank the reviewer for pointing us to this confusing structure.
With regards to the figure captions, we will make the following changes (additions are in bold) to the figure captions:

Fig. 3: "… MNDWI thresholds 0.1, 0.25, 0.4, 0.55, and 0.7 are marked (grey lines). **The sensitivity of the area, and to a lesser extent, the volume to changes in MNDWI threshold is low over a wide range of MNDWI values…**"

Fig. 5: " **Duration and accumulated precipitation of HPEs, and runoff coefficients of LFEs…** medium HPEs are marked with dots. **Both data sources show relatively good agreement for precipitation accumulation, but ERA5 is negatively biased compared with IMERG. Additionally, LFEs are only observed over the ten largest HPEs by magnitude. (b)**  **Effective runoff coefficient during LFEs computed by dividing the lake storage change with precipitation during LFE-generating HPEs.**"

Fig. 7: " synoptic-scale conditions during HPE5.1 (panels a and b), and HPE5.2 (panels c and d) **as an example for the southward intrusion of upper-level PV-features and a low-level cyclone, and a northward intrusion of moist air…**"

Fig. 8: **"Moisture dynamics during HPE5.1 and HPE5.2 (LFE5). The temporal evolution of the atmospheric column**  **is** interpolated to the catchment centre…**"**
Additionally, we will add at the end of the caption **"Precipitation evaporation and sublimation are evident before each of the two rain periods."**

Fig. 9: We will add at the end of the caption "**Moisture sources clearly differ between the two rain periods.**".

Fig. 10: We will add at the end of the caption "**Precipitation evaporation and sublimation are exhibited over two periods before the arrival of these air parcels into the catchment.**".

Fig. 11: We will add at the end of the caption "**Anomalies are clearly more emphasised during LFEs compared to HPEs and during strong HPEs compared to medium HPEs.**".

Furthermore, we would revise the structure of the manuscript such that all the parts dealing with the domino effect are concentrated near the end of the results section, as a "bridge" to the discussion.

In the results, evolution of HPE5 is given as an example for all HPE's. I think this is reasonable for the synoptic evolution. However, using the HPE5 as an example for the moisture sources isn't clear and the discussion about the proposed "domino effect" is misplaced. The discussion of the "domino effect" should be linked with fig 13, where it is clearest. In the way it is written, the mechanism is proposed before the full results are presented, which makes the claim weaker during the reading and stronger only after the full results are presented. I suggest moving section 4.2.2 to the discussion. In this way, you can lay out the full argument for the "domino effect" in one place, with all the key observations required to understand the mechanism already laid out.

We agree with the reviewer about the confusing aspect of the domino effect. Therefore, we have put all the observations that relates to it at the end of the results section, and we will further discuss it in the discussion.

Duration of the events seems to be the key component of the LFE, if the system persists for long enough there will be a LFE (Fig. 5a shows this clearly). This aspect of the system was not discussed much. What causes the system to persist for 5 days over 1 day?

We agree with the reviewer about the importance of the duration of events, and therefore added a short section in the discussion where we explain this: "**Persistence of upper-level conditions conducive to air lifting can happen, for example, when a strong, northward extending, blocking high is situated upwind in the central North Atlantic ocean.** An example of this is given by the November 2014 case study (Sect. 4.2) where PV structures appear repeatedly in the vicinity of the catchment **and a high persists over the central North Atlantic.** The persistent conditions allow for increased moisture transport, continuous formation of rain cells, and therefore also for a longer rainfall duration and areal coverage, which are needed to generate intense desert floods (e.g., Doswell et al., 1996; Morin and Yakir, 2014). These conditions lead to relatively long-duration HPEs, typically 3 d for the LFE-triggering HPEs." (L563-569).

Runoff coefficient is a very important aspect of the LFE's (as explained by the authors). However, it is unclear what generates the large Runoff coefficients. It would be beneficial to discuss this, even if there isn't a concrete answer to this problem.
We thank the reviewer for this suggestion and agree that this is indeed an important question. To answer it, we added a short description of what we suggest that could have been important in generating high runoff coefficient values. "**The higher runoff coefficient values are**

**presumably related to precipitation over the parts of the catchment where rocky surface are exposed in combination with higher rainfall intensities, as well as longer rainfall durations and shorter inter-event periods. All these aspects increase the runoff contributing area and its contribution to flood discharge (e.g., Rinat et al., 2021)."** (L599-602).

**Line comments:**

L11. Should be: eventuated
Corrected.

L116. Should be: spelled
Corrected.

L116. Add coordinates
Added.

Figure 1. Add location of cities for orientation: Bechar, Kerzaz, El Menia, etc.
Added.

Figure 1b. From the decrepitation and figure the properties of the lake are unclear. There is no need for the inset on the left, it is redundant with fig. 1a. Would be better to remove it and add a bigger image of the sebkha itself, and perhaps an image showing a time the lake was dry/drier, and a topographic transect showing the water level.
The figure is now revised (see below). We have removed the zoomed-out satellite map and added a cross section of the lake's bathymetry. We decided not to put a figure showing the empty lake, as it is barely visible in such an image.

[Figure]

L123. How deep does the water get? How deep does the water need to be for the lake to overflow?
The water gets to a bit less than 5 m when the lake is overflown. We have added this information to the text: "When the lake reaches full capacity, **~5 m depth and ~0.9 km³ (Fig. 1b and Fig. A2)**, which only happens in extremely rare floods…"

L126. How do you know what the geology is? You didn't cite a geological map.
We have now added a reference to a book chapter describing the geology of the region (Merzougui et al., 2017).

**L128.** Please add the location of these dams to figure 1, otherwise, these location names are meaningless. Do you take these dames into account when calculating the size of the catchment?

We have now added indications for the dams' locations on top of the location map (Fig. 1b). We do not take these dams into account for the size of the catchment, as water still flows downstream of the dams. The outflow of the Djorf Torba (the main dam) is reported in Sarra et al. (2023), however the raw data is unavailable to us, although we sent data requests to both the corresponding author of this paper and to the chief editor of the journal. Our observations, as well as satellite observations from September 2024 (not shown in the paper) suggest that the lake is still fed by large floods in the wadi, even after dams were constructed. We expect small events to be affected by the presence of the dams, however, as noted in the manuscript "… large HPEs are not expected to be heavily affected; the main dam in the catchment can only contain $\sim 0.25$ km$^3$ (Sarra et al., 2023), which is $\sim 4$ % of the mean catchment rainfall in the largest HPE (Fig. 5a)".

**L147.** Should be e.g., precipitation and evaporation? Evaporation of precipitation?

The latter is what we meant. Therefore, we rephrased into "evaporation of precipitation".

**L215.** Why not use model estimates of evaporation? e.g., the method used by: Zhou et al., NC (https://doi.org/10.1038/s41467-022-31125-6).

We appreciate the reviewer suggestion of directly calculating evaporation from the lake, which may result in more precise evaporation estimates during the times when the lake was full. However, we want to stress that we have addressed rough evaporation estimates mainly as a way to calculate potential errors to our method, as displayed in Fig. 5b. As this figure shows, major changes in the method used to estimate the evaporation rate can only affect the smallest LFEs (#4 and #6), while the effect over all other events would probably be negligible. Moreover, the paper by Zhou takes a global perspective, and therefore, unique properties of specific lakes are to some extent concealed by the large number of lakes. If we would like to use their model to directly calculate evaporation over the lake, we would need to assess a few more parameters which are not known to us. These include the water salinity (or activity), water temperature, the function coefficients a and b that depends on the local topography, stability conditions etc. When taking assumptions on these parameters we would introduce new errors in assessing the evaporation from the lake, and we believe the resulting values would not drastically differ from our current estimate of 20 cm per month, since the primary controls (air temperature, atmospheric saturation deficit, and wind) for evaporation are taken into account when giving this estimate.

**L263.** Where does the actual dq data come from? AR5? IMERG?

It is from ERA5. We will add this information to the text.

**L320.** 100 mm where? In how much time?

To make this clearer, we would rephrase into: "where average precipitation over the catchment reaches nearly 100\,mm during the event duration".

**L364.** Should be: predominantly

Corrected.

**L375.** It's not clear to me why you invoke this domino mechanism. If there is subsidence which prevents uplifting of the moisture, can't the moisture be caught in mid-levels and be pushed eastwards, which wouldn't require precipitation and evaporation? and the long residence time is because the moisture never reaches condensation levels?

As the reviewer suggests, subsidence can indeed cause moisture to be caught in mid-levels which would only trigger precipitation later, when this moisture is being uplifted, as suggested, e.g., by Rubin et al., 2007. However, if that would have been the case, the Sahara would not have been classified as a moisture source by the moisture source diagnostic (MSD).

Clearly long residence times of water vapour in the atmosphere also play a role in precipitation events in the Sahara (climatologically 6-10 days, see for example, Sodemann 2020). However, at this line in the text we discuss the results shown in Fig. 9c. These results were compiled based on the trajectory-based MSD that we present in the methods (Section 3.2.5). For this method to diagnose moisture uptake a physical process has to inject moisture into an air parcel. This can happen in different ways, e.g., through surface evaporation, turbulent mixing, as well as precipitation evaporation. At L. 375 we write that given the large share of uptakes happening in the Sahara, where surface evaporation is small or zero, another process is responsible. Given our results in Fig. 10 showing the trajectories arriving in the catchment during the first precipitation peak of LFE5 (HPE5.1) gaining humidity as they pass below a large-scale upper-level cloud system (see Fig. 10 and A5) we hypothesise that this is likely due to precipitation recycling. The fact that this water vapour is not lost to the surface but largely evaporated below the cloud is indeed relevant for precipitation formation further downstream in the catchment. The domino effect we propose is a mechanism of hand over of moisture between different atmospheric layers transporting air of different origin. We suggest that this mechanism is likely relevant for the transport of moisture deeper into the Sahara, which is only made possible through recycling of moisture from mid-level or upper-level cloud systems. We understand that in the current formulation this was not made clear enough. We will clarify this as follows (in bold we show new text):

"A large part of this precipitation evaporates or sublimates when falling into the relatively dry lower troposphere (possibly forming virgae). **Thereby a vertical connection between the upper-level clouds transporting moisture of tropical origin and the lower level dry airstream from the Sahel is established.** This moisture thus gets recycled and transported further north into the catchment in a domino-like process, which may repeat itself several times on the transport pathway from the tropics into the Sahara and finally into the catchment. **This recycling process likely contributes in destabilising the lower to mid troposphere, which together with the large-scale convergence**  of moisture from different tropical sources (tropical North Atlantic and from the Sahel, Fig. 8; similar observations were made by Knippertz et al., 2003) **and the forcing for ascent due to topography upon arrival in the catchment leads to the formation of a large-scale deep convective system forming intense precipitation.**"

It is also not clear why this is a prerequisite for the HPE event? If the source of the moisture is in the Atlantic and all that is happening is recycling within the troposphere, why do you suddenly get a downpour? I think the mechanism and its link to the HPE needs to be explained more clearly.

We have now revised the description of the domino effect, as reflected in our previous answers. However, it must be noted that the discussion about the moisture sources and the one about the triggering mechanism are two separate issues. We have added a short description of the need in such a mechanism at the beginning of the text dealing with it "...given the normally dry conditions over the surface of the Sahara, we can assume that mechanisms other than surface evaporation are involved in triggering such HPEs. Based on the evaporation of precipitation upwind (Fig. 10) and before the HPE starts (Fig. 8), we suggest that a pre-moistening of the atmosphere in the vicinity of the catchment is important for precipitation formation over Sebkha el Melah, and that recycling of moisture in a domino-like way throughout the upstream region within the Sahara is significant for HPEs. We term this moisture recycling mechanism "domino effect"."

With regards to Atlantic moisture, as seen in our climatology of moisture sources, it seems to play a major role, but not as big as Saharan moisture. In normal conditions, we assume that moisture from the Atlantic goes up the Atlas Mountains and is precipitated along the range. When it reaches the southeastern side of the range, a substantial portion of the moisture is already dried up because of the upwind rains, and therefore, only a small part is left for precipitation in the Sahara.

A downpour would occur only if enough moisture was transported into the region and is then combined with a strong enough lifting mechanism, as we describe in our discussion section.

L384 – 389 and Figure 8. At what height are you measuring RH? It is interesting the RH increases only when rainfall reaches the ground 22/11 at 3:00. If there is recycling of moisture before this, shouldn't RH go up already during the enhanced moisture input?
We thank the reviewer for pointing this missing information to us. The RH shown is at the surface (we will add the word "surface" into the caption), and therefore, heavily depends on rainfall but not on recycling of moisture further up in the atmosphere nor recycling of moisture further upstream along the transport pathway, as discussed in the domino effect.

Figure 10. What are the units of the x axis in a and b? If this is time, it should be noted in the figure or caption. Also, I'm not sure how this figure helps with depicting the domino effect (from lines 390-393).
It is indeed time (in h). We would add this information into the figure caption. With regards to the domino effect, we have revised the text describing it (see above). More specifically, this figure shows the location of the trajectories, which arrive in the layer producing precipitation that reaches the surface of the catchment at 6UTC on 23.11. In this figure we see the history of these air parcels forward in time. The vertical location of the air parcels during their 3D voyage through the atmosphere is shown in Fig. 10d geographical location of the air parcels in Fig. 10c. What we see in Fig. 10b is the vertical structure of the clouds and the content of specific humidity throughout the troposphere when we "sit" on the air parcels and look up and downwards to the surface. So, Fig. 10a,b show us vertical profiles at each geographical location of the air parcels of panel c. For each hourly time step between 13.11 and 23.11 we see a composite of the conditions along the vertical profiles at the location of the air parcels. It therefore gives us a mean atmospheric column along the airstream (consisting of the different air parcels that produce precipitation). This airstream is relatively consistent in its transport history, which is a prerequisite for performing such an analysis. We see the domino effect at play starting from about 72h before arrival until arrival in the catchment with the snow falling between 700 hPa and 400 hPa (pink contours in panel b, note the snow gets sublimated before reaching the melting level) and the increase in specific humidity first at about 500 hPa and then later on in the whole lower troposphere (panel a) as rain starts to form (blue contours panel b, note that the rain is not reaching the surface until arrival in the catchment).

Figure 13. This figure is essential to your arguments of the "domino effect", I think it should be presented earlier in the text. Is this based on the MSD analysis? Please explain how this data is constructed.
We have restructured the domino effect text. Fig. 13 is indeed tied to the MSD analysis. Therefore, we have added this information to the figure caption. Furthermore, we changed the text as follows: "To investigate the importance of the domino process in more detail we compiled a composite of the moisture sources of all medium HPEs, strong HPEs, and LFE-generating HPEs using our trajectory-based MSD results. Thereby we can show that the importance of moisture recycling through this domino process is likely considerable and intensifies with increased event magnitude. The origin of more than 60% of the moisture throughout all HPEs is classified as coming from the desert (Fig. 13: orange-yellow colours). While the largest moisture-contributing region to HPEs is the Sahara and the vicinity of the Sebkha el Melah catchment, during LFE-generating HPEs the role of the farther Sahara and even the Sahel is clearly more important than for strong and medium HPEs. During these non-lake-filling HPEs, the moisture originates from regions closer to the catchment and higher contributions from the Atlantic and the Mediterranean are observed."

We seriously thought about the suggestion of restructuring our order of results, but we don't agree that Fig. 13 would be better placed earlier in the manuscript. It is a composite over all events showing differences for different categories of extreme precipitation events and

therefore it seems more natural to us to first explain the mechanistic details of the domino process based on Figs. 9 and 10 and taking LFE5 as an illustrative example, and then summarize the results from all precipitation events to highlight the climatological importance of the suggested mechanism.

L482. What part of the tropics? How is this moisture getting to the catchment? How does it relate with the moisture coming in from the Atlantic? This statement seems to be new and requires a bit more clarification.

This description is related to the domino effect and was therefore revised. Moisture from equatorial Africa seems to precede the Atlantic moisture, however, it is being recycled along its way. In the second stage of the event (HPE #5.2) moisture from the Atlantic plays a more significant role (Fig. 9b). Our strategy is to explain the mechanism behind the domino effect early in the results section because it is an important aspect of effective moisture transport into the Sahara. Here we go one step further than in the case study of LFE 5 and generalise the importance of remote moisture and its recycling underways for producing large amounts of precipitation that are required for filling the Sebkha el Melah lake. Based on the MSD method we can unfortunately not say from which parts of the tropics the recycled moisture originally comes from. For this we would need numerical tracers such as we used in Dahinden et al. 2023 ASL or water isotope observations. This goes beyond the scope of this article but we will add a sentence mentioning this:

"However, to generate enough precipitation to fill the lake, moisture recycled from farther away is necessary, brought from the tropics and recycled over the Sahel and the Sahara. With the MSD approach used in this study it is not possible to identify the sources of the recycled moisture underway, however in a future study we plan to use a regional climate model with numerical tracers (as in Dahinden et al. 2023, which will allow to identify the tropical regions from which the moisture from the mid to upper-level cloud layers originate."

L608 – 614. I think the emphasis here is not precise. If during the mid-Holocene the lakes are higher, or more persistent, there are more LFE's and most likely an increase in the multiyear rainfall mean. So, the distinction you are making is not about the mean rainfall but about the type of system that causes the LFE. I think what you would like to say and are not saying clearly enough, is that given the modern conditions, a large scale northward shift of the African monsoon during the mid Holocene is not necessary. Rather, a higher tendency of westward cyclones could produce the same paleo archive. The question you should be raising is: given that models find it hard to move the monsoon this far north under the moderate forcing of the mid-Holocene, is it easier to produce more frequent cyclones under these conditions? If so, this would reduce the mismatch between paleo data and models.

We appreciate the reviewer's suggestion about the focus of this discussion part, and have therefore, added this suggestion to the discussion and mentioned it in the conclusions and abstract. We do, however, still think that the hypothesis we previously suggested is relevant and we have tried to make the text about it clearer in the revised version. A tendency towards more extreme weather events, with no change of the mean annual precipitation, would trigger a higher chance of getting the lake to be filled. When combined with the fact that the lake takes a few years to empty out, we think that this type of climate change can sustain generally higher lake levels in the Sahara.

**References**
Dahinden, F., Aemisegger, F., Wernli, H., and Pfahl, S.: Unravelling the transport of moisture into the Saharan Air Layer using passive tracers and isotopes, Atmos. Sci. Lett., 24, 1–11, https://doi.org/10.1002/asl.1187, 2023.

Merzougui, T., Mekkaoui, A., Mansour, H., and Graine-tazrout, K.: Hydrogeology of Béni Abbès: potential, hydrodynamics and influence on the palm field (Valley of Saoura, Algerian South-West), in: Aquifer Systems Management: Darcy's Legacy in a World of Impending Water Shortage, edited by: Chery, L. and de Marsily, G., Taylor & Francis, 269–278, 2007.

Sodemann, H.: Beyond turnover time: Constraining the lifetime distribution of water vapor from simple and complex approaches, J. Atmos. Sci., 77, 413–433, https://doi.org/10.1175/JAS-D-18-0336.1, 2020.

---

## Author Comment (AC2)

Reviewer #2

The paper "Meteorological ingredients of heavy precipitation and subsequent lake filling episodes in the northwestern Sahara" by Rieder et al. uses satellite observations and ERA5 output to to identify lake filling events in the northern Sahara and characterize the large-scale meteorological conditions associated with them. The manuscript is well written and addresses the important topic of heavy precipitation events in a complex part of the world and has interesting implications for paleoclimate. However, clarification of some key points are needed, so I recommend this paper be accepted with revisions.

We would like to thank Reviewer #2 for the time taken to read our manuscript and for the helpful comments. We are also glad to read the reviewer find the paleoclimatic implications interesting.
Please see below our detailed response to all the comments raised by the reviewer.

You've shown very clearly that the high precipitation events (HPEs) you've identified are associated with large-scale circulation patterns and moisture convergence, which makes sense. Are there conditions where the pattern of extratropical cyclones and and upper level PV anomalies *don't* result in these events? How frequently do these patterns coincide? It would be interesting to see how what composites of these large-scale patterns that do not produce HPEs look like: if they don't result in moisture convergence this would be a very strong support for your claim. Or if you showed cases of the low-level cyclone without the PV anomaly, that might show moisture transport but not the ascent. This is probably beyond the scope of your paper, but should be considered.

We agree with the reviewer that the co-occurrence/non-co-occurrence of upper-level PV features and surface cyclones is indeed an interesting question. To emphasise the importance of the upper level PV features, we show below (Fig. R1) the 315 K PV climatology for DJF for the years 2000-2021 for (a) all days, (b) days with LFE-generating HPEs,(c) strong HPEs, and (d) medium HPEs. This figure, as well as Fig. 12, Fig. A6-A7, highlight how anomalous upper-level and surface conditions are during HPEs. In the paper, we address the frequency of both surface cyclones (~5-10% of time; Wernli and Schwierz, 2006) as well as upper-level PV cutoffs (~5-10% of time; Portmann et al., 2021) in the vicinity of the catchment (L520-527). Of course, the fact that both systems have a similar frequency in the area may be a coincidence. However, there are many reasons to think that at least when precipitation peaks, this co-occurrence is not happening by chance. To demonstrate it, we took the liberty to use a figure from an earlier version of a preprint now accepted to Communications Earth and Environment by de Vries et al., showing not only the association of extreme precipitation with Rossby wave breaking (Fig. R2a; defined there as the occurrence of a PV streamer or a cutoff low) but also the precipitation surplus due to Rossby wave breaking. The surplus represent estimates of how much precipitation is enhanced or reduced due to Rossby wave breaking, and therefore, where positive (as in the studied catchment) precipitation is enhanced when cutoffs occur and is reduced when they are absent. That being said, in order to complete the answer of the non-co-occurrence of upper-level PV features and surface cyclones on HPEs in the region, further studies are needed. Therefore, we have added the following statement to our discussion: "Additionally, the presence of upper-level PV features and low-level cyclones does not warrant the formation of a HPE, and further studies are needed to determine the frequency of these features without HPE formation." (L527-528)

[Figure]

**Figure R1.** PV climatology at 315K for DJF months between 2000 and 2021 (a), during LFE-generating HPE days (b), strong HPEs (largest 20% of HPEs; c), and smaller HPEs (d).

[Figure]

**Figure R2.** Association of extreme precipitation and Rossby wave breaking (RWB; a), and the precipitation surplus due to RWB (b). The figure was adapted from de Vries et al., (accepted).

I don't fully understand why this moisture recycling "domino-effect" is required for these HPEs. I think this is an interesting hypothesis which should be explored further, but there isn't fully evidence for it. I would suggest moving this to a discussion instead of continually emphasizing it as something that is definitely happening. Could this possibly be due to choice of reanalysis? I always worry when I see a study that relies only on a single reanalysis, especially in regions that have sparse observations. If you used MERRA2, which might represent below cloud evaporation differently than ERA5, would you see the same regions of moisture supply?

We thank the reviewer for the opportunity to elaborate on the domino effect, as it seems the way we presented it in the manuscript made it unclear. To make the description of the domino effect clearer, we have now rearranged all the text that concerns it, and concentrated it in the last part of the results as a "bridge" to the discussion, where it is discussed further. The effect is described

in the manuscript in order to explain how it is possible for the Sahara and the dryland regions surrounding it to behave as moisture sources. Fig. 9a and c, and Fig. 13 show that these dry areas were marked by the moisture source diagnostic (MSD) as the origin of the moisture for precipitation during HPEs in the catchment. However, clearly, these areas cannot be a major primary source of moisture because they are dry at the surface. To explain this discrepancy, we analysed vertical profiles along the trajectories (Fig. 10) as well as the evolution of the vertical profile of the atmosphere over the catchment (Fig. 8). Both these figures hint towards evaporating precipitation that falls before (and upwind) the occurrence of the main HPE. Therefore, in the manuscript, we suggest that at least part of the moistening of the atmospheric column upwind of the catchment is caused by this type of precipitation evaporation, which explains why the MSD mark these areas as moisture sources.

As the reviewer suggests, different reanalysis products may yield different results in terms of the moisture diagnostic. However, since the synoptic scale flow patterns are rather similar between different reanalyses (e.g., Davies and Sprenger, 2024) and humidity in North Africa (as well as other parameters) is similar between MERRA2 and ERA5 (Baba et al., 2021; Johnston et al., 2021), we think that while the exact amount / location of this domino process can slightly differ between different data sources, the effect itself is not a consequence of the choice of reanalysis product. Clearly, redoing the whole analysis with another reanalysis dataset is beyond the scope of this manuscript, but it would indeed be interesting to assess the representation of this mechanism in different numerical models because the evaporation of precipitation below the cloud is a process that is not well constrained. As shown in a sensitivity study in Aemisegger et al. (2015) for a cold front passage case study over Europe using the regional numerical weather prediction model COSMO (Steppeler et al. 2003) neglecting below cloud evaporation locally can lead to an increase in rainfall intensity of 74%. In this study, we do not have the observational means to quantify the importance of this process, however, based on the detailed trajectory analysis shown in Fig. 10 we can descriptively pinpoint the mechanisms that lead to the handover of moisture from an upper-level ice cloud layer to a lower tropospheric air stream that is originally relatively dry and comes from the Sahel. What we present is indeed a phenomenological description: we diagnose a progressive increase in specific humidity (Fig. 10a) in the dry airstream from the Sahel as it travels just beneath the ice cloud from which the falling snow (Fig. 10b violet contour) gets fully sublimated as it falls in the airstream (trajectories' location is shown in Fig. 10a by the orange line).

Finally, we would like to emphasise that rather than the type of reanalysis used, we think that the assumptions and thresholds chosen in the setup of the moisture source diagnostics play a much more important role. From a current intercomparison effort of existing methods (comparing 20 ensembles from 7 different methods, Benedict et al., 2024) we see that the footprint areas are the same, while the weights of different contributions, in particular, the contribution of local vs. remote sources can differ between methods as well as between ensemble members of the same methods using different threshold settings. This is in agreement with earlier comparisons between Lagrangian and Eulerian moisture source analyses (Winschall et al. 2014).

One part of your methodology that confused me was the use of both LANDSAT and MODIS. Why don't you just use your algorithm on MODIS, instead of just using it for visual confirmation of these events? Some clarification on how you are using both of these and why would be greatly appreciated.

While our methodology is applicable to MODIS imagery, we chose to apply it to Landsat images because of the higher spatial resolution it gives. Landsat pixels are ~70 times smaller compared to MODIS's and therefore, provide much better estimates of the area covered with water over such a small lake, where high slopes characterise the edges of the lake. However, given the coarser temporal resolution of Landsat imagery, we used MODIS imagery to (a) verify the results obtained by our Landsat-based algorithm, and (b) to make sure we have successfully detected

all LFEs. For (b) we visually inspected the 10 largest HPEs, where we thought there might have been LFEs the algorithm missed (although eventually there were no such events).

**References**

Aemisegger, F., Spiegel, J. K., Pfahl, S., Sodemann, H., Eugster, W., and Wernli, H.: Isotope meteorology of cold front passages: A case study combining observations and modeling, Geophys. Res. Lett., 42, 5652–5660, https://doi.org/10.1002/2015GL063988, 2015.

Baba, M.W.; Boudhar, A.; Gascoin, S.; Hanich, L.; Marchane, A.; Chehbouni, A. Assessment of MERRA-2 and ERA5 to Model the Snow Water Equivalent in the High Atlas (1981–2019). Water 2021, 13, 890. https://doi.org/10.3390/w13070890

Benedict, I., Weijenborg, C., van der Ent, R., Keune, J., Koren, G., and Kalverla, P.: A moisture tracking intercomparison study - Addressing the uncertainty in modelling the origins of precipitation, EMS Annual Meeting 2024, Barcelona, Spain, 1–6 Sep 2024, EMS2024-1040, https://doi.org/10.5194/ems2024-1040, 2024.

Davies, H. C., & Sprenger, M. (2024). Cyclone-like features within the stratospheric polar-night vortex. Geophysical Research Letters, 51, e2024GL109529. https://doi.org/10.1029/2024GL109529

Johnston, B.R.; Randel, W.J.; Sjoberg, J.P. Evaluation of Tropospheric Moisture Characteristics Among COSMIC-2, ERA5 and MERRA-2 in the Tropics and Subtropics. Remote Sens. 2021, 13, 880. https://doi.org/10.3390/rs13050880

Rubin, S., Ziv, B., and Paldor, N.: Tropical Plumes over Eastern North Africa as a Source of Rain in the Middle East, Mon. Weather Rev., 135, 4135–4148, https://doi.org/10.1175/2007MWR1919.1, 2007.

de Vries, A., J., Armon, M., Klingmüller, K., Portmann, R., Röthlisberger, M., and Domeisen, D. I. V.: Breaking Rossby waves drive extreme precipitation in the world's arid regions, Accepted to publication in Communications Earth and Environment.

Winschall, A., Pfahl, S., Sodemann, H., and Wernli, H.: Comparison of Eulerian and Lagrangian moisture source diagnostics – the flood event in eastern Europe in May 2010, Atmos. Chem. Phys., 14, 6605–6619, https://doi.org/10.5194/acp-14-6605-2014, 2014.